# When and how can inexact generative models still sample from the data manifold?

**Nisha Chandramoorthy**
Department of Statistics
Committee on Computational and Applied Mathematics
The University of Chicago
Chicago, IL 60637
nishac@uchicago.edu

**Adriaan de Clercq**
Department of Statistics
Committee on Computational and Applied Mathematics
The University of Chicago
Chicago, IL 60637

## Abstract

A curious phenomenon observed in some dynamical generative models is the following: despite learning errors in the score function or the drift vector field, the generated samples appear to shift *along* the support of the data distribution but not *away* from it. In this work, we investigate this phenomenon of *robustness of the support* by taking a dynamical systems approach on the generating stochastic/deterministic process. Our perturbation analysis of the probability flow reveals that infinitesimal learning errors cause the predicted density to be different from the target density only on the data manifold for a wide class of generative models. Further, what is the dynamical mechanism that leads to the robustness of the support? We show that the alignment of the top Lyapunov vectors (most sensitive infinitesimal perturbation directions) with the tangent spaces along the boundary of the data manifold leads to robustness and prove a sufficient condition on the dynamics of the generating process to achieve this alignment. Moreover, the alignment condition is efficient to compute and, in practice, for robust generative models, automatically leads to accurate estimates of the tangent bundle of the data manifold. Using a finite-time linear perturbation analysis on samples paths as well as probability flows, our work complements and extends existing works on obtaining theoretical guarantees for generative models from a stochastic analysis, statistical learning and uncertainty quantification points of view. Our results apply across different dynamical generative models, such as conditional flow-matching and score-based generative models, and for different target distributions that may or may not satisfy the manifold hypothesis.

## 1   Introduction

Given samples from a *target* distribution, a generative model (GM) outputs more samples (approximately) from the target. Most generative models accomplish this task using a dynamical formulation of probabilistic measure transport. They solve an optimization problem for a vector field or a *drift* term of a deterministic or stochastic process that produces the desired target samples in finite time. Flows under the learned vector field transports probability densities between some source density

39th Conference on Neural Information Processing Systems (NeurIPS 2025).

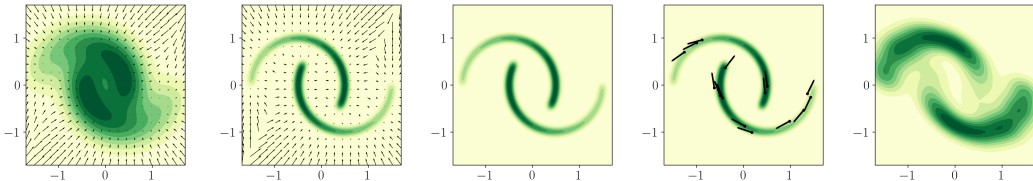

Figure 1: Robustness of the support under perturbations of SGMs. Columns 1 and 2: score vector field (lines) and the density (contours) near the start time and toward the end respectively. Notice that the score field is nearly orthogonal to the target support. Column 3: generated target density. Column 4: Leading finite-time Lyapunov vectors at the end, which are noticeably aligned with the target support. Column 5: kernel density estimate of the distribution generated by a process corrupted by *large* errors in the score. The density is shifted primarily tangent to the data manifold.

and the target. For instance, in diffusion models or score-based generative models (SGMs) and their variants [52, 51, 11], the vector field corresponds to a score of a noising process (e.g., an Orstein-Uhlenbeck process starting initialized with the given target samples) and the source density is that achieved at the end of the noising process. On the other hand, in conditional flow matching variants [37, 56, 57, 55], stochastic interpolants and Schrödinger bridge-based variants [18, 3, 2] and using Neural ODEs [61, 14], the learned vector field is constructed by specifying a path (e.g., straight line path or optimal transport paths) on sample space, between samples according to a fixed source density and the given target samples. In every case, there are inevitably errors incurred in learning the vector field as a neural network for multiple reasons, including optimization errors, approximation errors, discretization errors during time integration (of the underlying deterministic or stochastic process) and finite sample errors. These errors will propagate through the generating process and therefore be reflected in the predicted target samples. However, when are the predicted target samples close to high density regions of the target? How can we formalize this error propagation? Further, how can we determine the robustness of a given GM to these errors in the vector field? In this work, we provide answers to these questions by taking a dynamical systems approach to GMs. Even as the use of GMs proliferates (see e.g., [60] for a review), their utility in critical applications, e.g., in climate predictions [40] especially hinges on theoretical guarantees for their practical implementations. Significant research progress has been achieved in obtaining convergence guarantees (see e.g., [17, 34]; section 5) for SGM-variants under minimal assumptions on the target as learning errors in the score vanish. But rather than convergence to the distribution, a property that may be more pertinent to practical applications across engineering and data science is the *robustness of support*, which is the subject of this work. In other words, since learning errors are inevitable, a characterization of when a generative model will still be able to capture the high-density regions, or produce *physically relevant* or plausible samples, would be enormously useful.

Our definition of robustness is motivated by many empirical and theoretical studies on the feature learning and generalization of SGMs under the manifold hypothesis [47, 13]. Figure 1 depicts these observations for a two-dimensional two-moons target density (supported almost on a 1D curve), generated by an SGM using analytical scores, shown in the third column. When at each step of the generating process, we make a deterministic error in the score estimation, we observe that the generated distribution shifts along the support, as illustrated in Figure 1 (rightmost column). Surprisingly, the errors in the score estimation, even when large, do not cause the generated samples to move to zero probability regions of the target, which are off the moons for the target shown in Figure 1. A natural question that arises is when such a robustness of the support may be expected. To answer this question, we formally analyze this behavior of the time-inhomogeneous Markov chain that generates the samples as a random dynamical system. As a byproduct of our analysis, we characterize precisely how geometric information about the support is extracted by robust generative models. Our main contributions are as follows:

**Robustness of support.** Under mild regularity assumptions on the generating dynamics, we show that an infinitesimal change in the predicted target is only supported where the target density is supported as the perturbation size tends to 0. This explains our observed robustness of support even in high-dimensional data distributions (see Appendix G.4 for response to score perturbations on the CIFAR10 distribution).

**Alignment.** When the leading finite-time Lyapunov vectors (LVs; see section 4 for precise definitions), which represent the principal directions of deformation of the sample space at the final time, have a negligible component normal to the target support, we show that the generating process learns the support. This is evident in Figure 1(column 4), where the leading LVs (shown as black lines) are tangent to the support, and robustness of the support holds.

**Dynamical mechanism for alignment.** We prove sufficient conditions that characterize when the generating dynamics has LVs that align with the support of the data distribution. One sufficient condition, intuitively, turns out to be that the vector field acts as an attracting force to the support, which is satisfied for the SGM dynamics in Figure 1 (columns 1 and 2).

**Practical implications.** Note that there are many efficient, numerically stable algorithms, which can be implemented with automatic differentiation, to compute these finite-time Lyapunov vectors. Thus, we can efficiently obtain the tangent spaces to the data manifold using any *aligned* GM. Finally, we show that if a GM has the alignment property, so does the same GM under learning errors. Since our analyses do not assume a specific dynamics, our results apply to all practical implementations (e.g., using predictor-corrector integration or ODE integration) and across different dynamical generative models (e.g., SGMs or conditional flow matching).

## 2 Preliminaries: generative models as random dynamical systems

Let $p_{\text{data}}$ be the data (target) distribution with a compact support $M \subset \mathbb{R}^D$. The distribution $p_{\text{data}}$ is singular under the manifold hypothesis, i.e., $\dim(M) = d < D$. Let $\rho_0$ be a given source probability density in $\mathbb{R}^D$. Fixing $\rho_0$ and a density that approximates $p_{\text{data}}$, there can be infinitely many *couplings* between them. That is, for many sequences of functions $\{F_t\}_{0 \leq t \leq \tau}$, sample paths $X_t = F_t(X_{t-1})$ can have the same starting and ending distributions, i.e., $X_0 \sim \rho_0$ and $X_\tau \sim p_{\text{data}}$. What enables a given sequence $\{F_t\}$ to have the robustness of support property? To study this question, we treat the sequence $F_t$ as a (random) dynamical system.

In score generative models or diffusion models [52], the dynamics, $F_t$, on path space is the stochastic reverse process. At each time $t = 0, 1, \cdots, \tau - 1$, the map $F_t^\Xi$ on sample space represents the dynamics under an instance of the *noise* path, denoted by $\Xi$. This noise path is a sequence $\Xi = \{\xi_0, \cdots, \xi_{\tau-1}\}$ of independent Gaussian random variables, which must be viewed as a (scaled) time discretization of a Brownian motion. Since, with a known score, a deterministic, time-dependent process (time-integrated solution of a probability flow ODE) can effect the same dynamics on probability space, for simplicity, we consider deterministic, non-autonomous maps $F_t$ throughout. However, note that the perturbation analysis both on probability space and tangent space in the remainder of this paper applies pathwise (for each $\Xi$) (see [5, 29] for a rigorous treatment of Lyapunov vectors and exponents for random dynamical systems), even if we consider $F_t$ as a stochastic process.

In a dynamical formulation of a GM, $F_t$ is typically partially represented by a neural network. In general, $F_t(x) = x + \delta t\, v_t(x)$, where the drift vector field $v_t$ (a neural network) is learned using samples from $p_{\text{data}}$ and $\delta t$ is a small timestep. For instance, in a score-based generative model based on an Ornstein-Uhlenbeck noising (forward) process, $F_t$, the dynamical system representing the reverse process can be written as $v_t(x) = \theta\, x + \sigma^2 s_t(x), t \leq \tau$, where the score functions of the OU process, $s_t = \nabla \log \rho_t$, are represented using a neural network. With no learning errors, we may write that $F_\sharp^\tau \rho_0 = \rho_\tau$, where the pushforward notation ($\sharp$) for probability densities means the following: $T_\sharp \rho = \pi$ implies that if $x \sim \rho$, $T(x) \sim \pi$. Explicitly, the pushforward operator is a linear operator on densities, and in the context of dynamical systems, called the Frobenius-Perron or the transfer operator: $\mathcal{L}_t \rho := \rho \circ F_t^{-1}/|\det dF_t| \circ F_t^{-1}$, assuming that $F_t \in \mathcal{C}^1(\mathbb{R}^D)$. We use exponential notation, $F^{\tau+1} := F_\tau \circ F^\tau$, with $F^0 = \text{Id}$, to denote $\tau$ iterations of the process.

**Tangent space.** From here on, we will be considering the evolution of infinitesimal perturbations along sample paths of $F_t$. These will represent small learning errors that will perturb the evolution of sample paths. Roughly speaking, at each point $x \in \mathbb{R}^D$, the directions of infinitesimal perturbation are also in $\mathbb{R}^D$. More precisely, at each point $x$, we have a tangent space (isomorphic to $\mathbb{R}^D$) denoted by $T_x \mathbb{R}^D$ that represents all possible infinitesimal perturbations or *tangent vectors* at $x$. A vector field, say $v : \mathbb{R}^D \to T\mathbb{R}^D$, is a function that maps a base point $x \in \mathbb{R}^D$ to a tangent vector $v(x)$ in $T_x \mathbb{R}^D$. We refer the reader to Appendix C for further background on tangent spaces.

# 3 Why GMs sample along the support even under learning errors

In this section alone, we assume that $p_{\text{data}}$ is absolutely continuous (i.e., $d = D$) and its density is $\rho_\tau$: i.e., the generative model $F^\tau$ is exact/convergent (ignoring errors on $\mathcal{O}(\delta t)$). Consider now the perturbed dynamical system, $F_{t,\epsilon}(x) = x + \delta t\, v_t(x) + \epsilon\, \chi_t(x)$, where the vector field $\chi_t$ represents a time-dependent perturbation, and the original model $F_t$ is recovered with $\epsilon = 0$. It is important to note that *tangent perturbation* does *not* mean perturbation along the tangent space $T_x M$, but rather just a perturbation (a tangent vector) in $T_x \mathbb{R}^D$. In other words, the perturbation $\chi_t(x)$ at an $x$ can be in any direction in $\mathbb{R}^D$.

Perturbations to the dynamics here are models for statistical errors in the learned drift, $v_t$. That is, we define a parameterized family of maps $\epsilon \to F_{t,\epsilon}$ at each time $t$, with $F_\epsilon^t = F_{t-1,\epsilon} \circ F_\epsilon^{t-1}$. The corresponding dynamics on probability densities is given by $\rho_{t,\epsilon} = \mathcal{L}_{t,\epsilon}\rho_{t-1,\epsilon}$, with the corresponding Frobenius-Perron/transfer operator, $\mathcal{L}_{t,\epsilon}$. We omit the subscript $\epsilon$ to refer to the unperturbed system, $\epsilon = 0$. For instance, $\mathcal{L}_{t,\epsilon}$ and $\mathcal{L}_t$ are respectively the Frobenius-Perron operator at time $t$ of the perturbed and unperturbed systems. Using exponential notation for $\mathcal{L}_{t,\epsilon}$ as well, fixing the source density $\rho_0$ we write the perturbed density, $\rho_{\tau,\epsilon} = \mathcal{L}_\epsilon^\tau \rho_0$.

Recall that for a vector field $u$, and a differentiable scalar function $f$, its directional derivative is given by the scalar field defined as $u(f)(x) := \lim_{\epsilon \to 0}(f(x + \epsilon u(x)) - f(x))/\epsilon$. Now let $u$ be a vector field such that the directional derivative of $\rho_\tau$ with respect to $u$, denoted by $u(\rho_\tau)$, is the perturbation in the predicted density due to the $\epsilon$ perturbation of the dynamics. That is, let the vector field $u$ be such that $u(\rho_\tau) = \partial_\epsilon|_{\epsilon=0}\mathcal{L}_\epsilon^\tau \rho_0$. Intuitively, the derivative $\partial_\epsilon|_{\epsilon=0}\mathcal{L}_\epsilon^\tau$ gives the *statistical response* after time $\tau$ of the dynamics $\{F_\epsilon^t\}$. We refer to $u$ as the target response field, since it is a vector field that measures the direction and magnitude of the perturbation to the target density. We now understand the target response field, $u$, through explicit expressions for the statistical response operator, $\partial_\epsilon|_{\epsilon=0}\mathcal{L}_\epsilon^\tau$.

For a given test function $f : M \to \mathbb{R}$, $\mathbb{E}_{x \sim \rho_\tau} f(x) := \langle f, \rho_\tau \rangle = \langle f, \mathcal{L}^\tau \rho_0 \rangle = \langle f \circ F^\tau, \rho_0 \rangle$, from a change-of-variables in the integration (one can view this as a conservation principle for probability mass). Here, $\langle \cdot, \cdot \rangle$ refers to an $L^2$ inner product with respect to Lebesgue measure on $\mathbb{R}^D$. Now for any observable $f$, we define its time-dependent response, $r_t(f)$, to be a scalar field that denotes the change in its value along a path due to the perturbation to the dynamics. More precisely, let $r_t(f) := \lim_{\epsilon \to 0}(1/\epsilon)\left(f \circ F_\epsilon^t - f \circ F^t\right)$, so that, by taking adjoints, we have $\langle f, \partial_\epsilon|_{\epsilon=0}\mathcal{L}_\epsilon^\tau \rho_0 \rangle = \langle f, u(\rho^\tau) \rangle = \langle r_\tau(f), \rho_0 \rangle$. Now, using the definition of $r_t(f)$, for any $f$ and $t$, we can derive the following recursive relationship,

$$r_t(f) = \lim_{\epsilon \to 0} \frac{1}{\epsilon}\left[\left(f \circ F_\epsilon^t - f \circ F_{t-1,\epsilon} \circ F^{t-1}\right) + \left(f \circ F_{t-1,\epsilon} \circ F^{t-1} - f \circ F^t\right)\right]$$
$$= r_{t-1}(f \circ F_{t-1}) + (\nabla f \circ F_{t-1} \cdot \chi_{t-1}) \circ F^{t-1}. \tag{1}$$

Here, we have used the definition of the perturbation vector field $\chi_t$ at time $t$ by $\chi_t(x) := \lim_{\epsilon \to 0}(1/\epsilon)(F_{t,\epsilon}(x) - F_t(x))$, which isolates the error made due to the perturbation to the dynamics at time $t$. Since both $F^0$ and $F_\epsilon^0$ are the identity, we have $r_0(f)$ to be the zero function for all $f$. Applying the recursive relationship (1), we obtain $r_t(f) = \sum_{i=0}^{t-1}\left(\nabla(f \circ F_{t-1} \circ \cdots \circ F_{i+1}) \circ F_i \cdot \chi_i\right) \circ F^i$. Substituting this explicit expression for the response of $f$, we obtain the following expression for the statistical response, $\langle f, \partial_\epsilon|_{\epsilon=0}\mathcal{L}^\tau \rho_0 \rangle = \langle f, u(\rho_\tau) \rangle = \langle r_\tau(f), \rho_0 \rangle = \sum_{i=0}^{\tau-1}\langle \nabla(f \circ F_{T-1} \circ \cdots \circ F_{i+1}) \circ F^{i+1} \cdot \chi_i \circ F^i, \rho_0 \rangle$. Upon integration by parts and change of variables, we obtain for test functions $f$ that vanish on the boundary $\partial M$,

$$u(\rho_\tau)(x) = -\rho_\tau(x) \sum_{i=0}^{\tau-1}(\text{div}(\chi_i) + \chi_i \cdot s_i) \circ F_i^{-1} \circ \cdots \circ F_{\tau-1}^{-1}(x). \tag{2}$$

The above expression indicates that the statistical response is the product of the target density, $\rho_\tau$, with a scalar field. In other words, when $p_{\text{data}}$ is non-singular and $F_t$ is continuously differentiable on $\mathbb{R}^D$, we obtain that the statistical response is supported only where the target density is non-zero, in the limit $\epsilon \to 0$. See Appendix B for an extension of the above computation to stochastic processes and singular $p_{\text{data}}$.

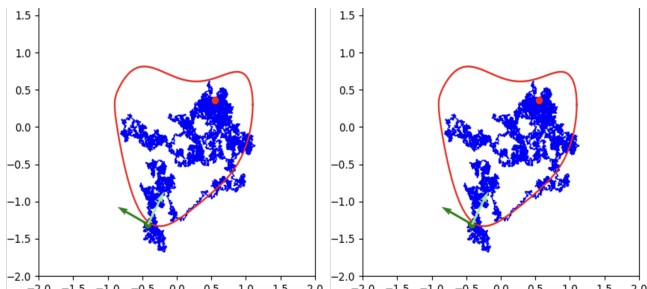

Figure 2: The finite-time Lyapunov vectors shown in green at two samples of the predicted distribution. The sample path of a diffusion model starting at the red dot is shown in blue. The leading backward Lyapunov vector (BLV), shown in a darker green, shows alignment with the data manifold. See Appendix G.2 for more details on this two-dimensional example.

## 4 A dynamical mechanism for robustness

In the previous section, we looked at statistical response to learning errors for non-singular targets. Here, we propose a mechanism for the robustness of the statistical response by looking at infinitesimal perturbations along sample paths (dynamics on the tangent bundle) for both singular and non-singular target distributions. To do this, we carry out a pathwise analysis that elucidates how learning errors propagate when we apply the dynamics. This is an adaptation of linear perturbation theory from dynamical systems theory (see [28] for a textbook exposition), and we illustrate the required concepts below.

Fixing a sample path, we deduce sufficient conditions on the stability of the sample path that is consistent with the robustness of support. To this end, we must define a notion of sample path stability. We say that a path initialized with $x_0 \sim \rho_0$ is *robustness-consistent* if at its end point, $x_\tau$, the directions of accumulated infinitesimal perturbations point along the data manifold and not away from it. To explain the reasoning behind this notion and define it more precisely, we provide a brief primer on tangent dynamics.

**Dynamics on tangent space.** Fix a sample path $x_t = F_t(x_{t-1})$. Now, consider applying an infinitesimal perturbation (similar to in section 3) at time 0. Then, after one step, $F_\epsilon^1(x_0) = F^1(x_0) + \epsilon dF_0(x_0) \chi_0(x_0) + \mathcal{O}(\epsilon^2)$. Hence, after one step, in the limit $\epsilon \to 0$, we have $(F_\epsilon^1(x_0) - x_1)/\epsilon = dF_0(x_0) \chi_0(x_0) =: \chi_1(x_1)$, which we may define as a new vector field $\chi_1$ evaluated at the point $x_1$ (a tangent vector at $x_1$). Iterating this definition, we can define a linear dynamical system, $dF^t = dF_{t-1} \circ F^{t-1} \cdots dF_0$ that acts on the tangent bundle and evolves an infinitesimal perturbation along a vector field applied at time 0 to the corresponding vector field at time $t$. Considering these linear evolutions, we can define the principal directions of deformation (the most sensitive directions of perturbation) by considering the leading eigenvectors of $(dF^t)(dF^{t\top})$, which we will call the finite-time Lyapunov vectors. The eigenvalues are generally arranged in decreasing order, with at least one positive eigenvalue indicating expanding directions or diverging infinitesimal perturbations. The time-asymptotic properties of these matrix are well-studied via the classical Furstenberg and Oseledets ergodic theorems (see [5] for a textbook exposition). However, we are only concerned with finite-time dynamics here, since GMs considered here are only defined for $0 \le t \le \tau$.

By definition, the least stable Lyapunov subspace is aligned with the most sensitive directions of (infinitesimal) perturbations in the dynamics. In Figure 2, we illustrate the two Lyapunov vectors (two one-dimensional Lyapunov subspaces) computed along a two-dimensional sample path (in blue). Here the data manifold (the support of the data distribution) is a one-dimensional curve shown in red. For more details on this experiment, including $p_{\text{data}}$, which is singular here, we refer the reader to Appendix G.2. The leading/least stable Lyapunov vector is shown in green, and interestingly, it appears tangent to the data manifold. Intuitively, this means that the direction where most errors will result along the sample path is aligned with the data manifold. In other words, the direction of the most stable Lyapunov vector, where any error made along the sample path decays quickest, is away from the data manifold. This means that the sample path is attracted to the data manifold, even under (small) errors. We formalize this intuition in the proposition below for convergent generative models [34].

For now, assume that the support of $p_{\text{data}}$ is a compact $d$-dimensional subset of $\mathbb{R}^D$, which $d < D$. On the boundary of the support of the data (target) distribution, intuitively, the score is orthogonal to the support. This is because $p_{\text{data}}$ is zero outside of the data manifold, and non-zero on it; thus the gradient log or score points in the direction of this sharp change in $p_{\text{data}}$. We will define robustness-consistency as the condition where the least stable $d$-dimensional Lyapunov subspace is orthogonal to the score. When the target score has a small (or zero) component along the least stable Lyapunov subspace, this means that the tangent directions most sensitive to perturbations are along the data manifold (as in the example in Figure 2). In this section, we give sufficient conditions for the robustness-consistency of sample paths. We will show how this improves our qualitative understanding of why some generative models are robust. Moreover, we will provide a computable criterion for verifying the robustness of a given generative model. Before this, we first describe the most sensitive subspaces or constructive approximations of the least stable Lyapunov subspaces of the tangent spaces associated with sample paths.

**Most sensitive subspaces.** Let $\{x_t\}$ be a fixed sample path (trajectory) of the generative model $F^\tau$. When we have an exact generative model, if $x_0 \sim \rho_0$, then $x_\tau \sim p_{\text{data}}$. Recall that the set $M \in \mathbb{R}^D$ is the support of $p_{\text{data}}$, and the effective dimension of the support is $d <= D$. We define $E_0^d$ to be a randomly chosen $d$ dimensional subbundle (a $D \times d$ orthogonal matrix at each $x$). We can consider the orthogonal decomposition of $T_{x_0}M$ (which is isomorphic to $\mathbb{R}^D$) to be $T_{x_0}M = E_0^d(x_0) \oplus E_0^{d\perp}(x_0)$. Now, the evolution of these subspaces under the time-dependent Jacobian matrix field, $dF_t$, gives the tangent dynamics: $dF_t \, E_t^d = E_{t+1}^d \, R_t$. By construction, since $E_{t+1}^d(x)$ at each $x$ is orthogonal, the above tangent equation is simply a QR decomposition of the $D \times d$ matrix $dF_t \, E_t^d$. The diagonal elements of $R_t$ represent stretching or contraction factors under the linearization $dF_t$. We refer to $E_t^d$ constructed this way as the most sensitive subbundle because as $t \to \infty$, this converges to the top $d$-dimensional Oseledets subbundle (backward Lyapunov vector bundle). See Appendix A.

**Proposition 4.1.** *[Predicting the support with convergent and aligned generative models.] For any $\delta > 0$, let $\epsilon_0 > 0$ be such that a convergent generative model (see Appendix D), $F^\tau$, produces $n$ samples, $\{y_i\}, i \in [n]$, such that with probability $\geq 1 - \delta/2$ over the generated samples, $\|T(y_i) - y_i\| \leq \epsilon_0$ and $T(y_i) \sim p_{\text{data}}$. Additionally, let $F^\tau$ be such that the most sensitive $d$-dimensional subspace of the tangent space, $E_\tau^d(x)$, spans $T_x \partial M$ with probability $\geq 1 - \delta/2$. Then, for some $c > 0$, there exists a binary function $f : \mathbb{R}^D \to \{+1, -1\}$ such that, $\mathbb{E}_{x \sim p_{\text{data}}} \mathbb{1}_{f < 0}(x) \leq cn^{-1} \log n + n^{-1} \log(n^2/2\delta)$, with probability $\geq 1 - \delta$.*

*Proof.* (Sketch) Given $n$ predicted samples, $y_i$, let $x_i = T(y_i) \sim p_{\text{data}}$ be samples from $p_{\text{data}}$ obtained by applying the $L^2$-optimal transport map. Let $f(x) = \text{sgn}(w \cdot \Phi(x) + b)$ be a one-class kernel-based classifier trained on $x_i, i \in [n]$. Then, $f$ satisfies the margin-based generalization bounds (see e.g., Theorem 3 of [58]; [49]) in the statement of the proposition. By assumption, with probability $\geq 1 - \delta/2$, we have, since the samples $y_i$ are of the form, $y_i = x_i + \epsilon_0 v_i$, where $v_i \in T_x \partial M$, the predictions $f(y_i) = f(x_i)$ and therefore the confidence margin of $f$ does not change. Taking union bound, we obtain the result, with $c = \mathcal{O}(1/\text{margin})$. See Appendix D for an elaboration. $\square$

The above lemma therefore says that the alignment with the data manifold of the least stable directions is crucial for robustness of the support, as defined in section 3. We formally define this alignment as follows.

**Definition 4.2.** We say that a generative model $F^\tau$ has the alignment property if, $E_\tau^d(x)$, the top $d$-dimensional Lyapunov subspace (spanned by the top $d$ least stable Lyapunov vectors) is orthogonal to the target score, $s_\tau(x)$, at each $x \in \partial M$, the boundary of the support of $\rho_\tau$.

First, we provide some intuition for the definition of alignment. We claim that a robust predicted distribution (in the sense of robustness of the support) has a highly anisotropic score near the data manifold (the true support of the target). More precisely, for a robust distribution, the score components along the data manifold are smaller than the components normal to it. Thus, it is reasonable to conclude that when a robust distribution shows such an anisotropicity along $E_\tau^d \oplus E_\tau^{d\perp}$, the most sensitive directions $E_\tau^d$ are aligned with (the tangent bundle along) the data manifold. Before we state our main result, which provides a sufficient characterization of generative models that have the alignment property, we define anisotropic derivatives.

**Tangent and normal derivatives.** We consider the orthogonal decomposition of the vector field (drift term) $v_t$ on the tangent bundle $T\mathbb{R}^D = E_t^d \oplus E_t^{d\perp}$. That is, we write $v_t(x) = E_t^d(x)v_{t,d}(x) +$

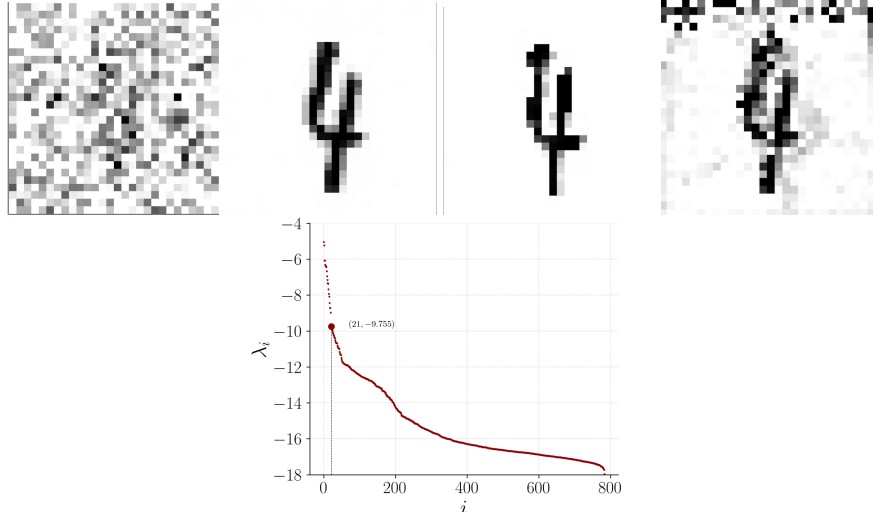

Figure 3: Top row (Column 1): noise image from source distribution. (Column 2): an MNIST [19] digit generated by a score-based generative model. See Appendix G for training details. (Column 3): generated image corrupted by the most sensitive Lyapunov vector. See Appendix A for details on the computation of these vectors. (Column 4): generated image corrupted by the 100th most sensitive LV at the same noise level. This shows that moving along very stable LVs, for indices greater than the intrinsic dimension, results in leaving the data manifold. Bottom row: finite time Lyapunov exponents associated with the generating process in an SGM, with a small gap at an index close to the intrinsic dimension, signifying superstability off the data manifold.

$E_t^{d\perp}(x)v_{t,d\perp}(x)$, defining the components, $v_{t,d}(x) = E_t^{d\top}(x)v_t(x) \in \mathbb{R}^d$ and $v_{t,d\perp}(x) = E_t^{d\perp\top}(x)v_t(x) \in \mathbb{R}^{D-d}$ of the vector $v_t(x)$. Let $\nabla_{t,d}$ indicate the differential "along the subspace $E_t^d$,", i.e., $\nabla_{t,d}f(x) = [\lim_{\epsilon\to 0}(f(x+\epsilon e_{t,d}^1(x))-f(x))/\epsilon, \cdots, \lim_{\epsilon\to 0}(f(x+\epsilon e_{t,d}^d(x))-f(x))/\epsilon]$, where $e_{t,d}^i(x)$ is the $i$th column of $E_t^d$. The differential, $\nabla_{t,d\perp}$ is defined similarly as the directional derivatives along the subspace $E_t^{d\perp}$. Finally, we also define the contraction factors $\alpha_t := \|R_t\|_\infty$ and $\beta_t := \|dF_t E_t^{d\perp}\|_\infty$, where the $\infty$ norm over the scalar field is the supremum over $x \in M$. By definition, $1 \ge \alpha_t \ge \beta_t$, at all $t$.

**Theorem 4.3.** *[Alignment of generative models.]* *Let $M \subset \mathbb{R}^D$ be the compact set where the target distribution $p_{\text{data}}$ is supported. When the support $M$ of the target is a $d$-dimensional manifold with $d \le D$, for an exact generative model that is compressive, the alignment property holds under the following conditions: for some $\delta \in (0,1)$, there exists a time $t^* \in [0,\tau]$ such that i) $\prod_{t\le n\le\tau} \alpha_n \ge (1+\delta)^{\tau-t}, t \ge t^*$; ii) $\|(\text{Id} + \delta t\, \nabla_{t,d}v_{t,d})^{-1}\nabla_{t,d}^2 v_{t,d}\|_\infty \le \delta^{1/(t-\tau)}$, for all $t \in [t^*,\tau]$; and, iii) the cross derivatives $\|\nabla_{t,d\perp}v_{t,d}\|, \|\nabla_{t,d}v_{t,d\perp}\| \approx 0$, for $t \in [t^*,\tau]$. Intuitively this means that the generating dynamics may include an expansive phase and the vector field $v_t$ is a uniform "attracting force" at the end.*

*Proof.* Recall the covariance of $E_t^d$ in the sense that $dF_t E_t^d = E_{t+1}^d R_t$, where $R_t$ is an upper triangular matrix in $d$ dimensions. At each time $t$, for some $\delta_t > 0$, we choose local coordinates on a $\delta_t$-neighborhood of $x_t$ such that $x_t$ maps to 0 in $\mathbb{R}^D$ and $E_t^d$ maps to the standard basis of $\mathbb{R}^d$ under the differential. If $M$ is a smooth embedding in $\mathbb{R}^D$, we may define derivatives of all orders of these local coordinates. We now analyze the behavior of the Jacobian $dF_t$ and the score vector field, $s_t$ in the two distinct phases of the dynamics: for $t < t^*$ and at the end, when $t > t^*$. In local coordinates, we have, $dF_t(x) = \begin{bmatrix} \text{Id} + \delta t\, \nabla_{t,d}v_{t,d}(x) & \delta t\, \nabla_{t,d\perp}v_{t,d}(x) \\ \delta t\, \nabla_{t,d}v_{t,d\perp}(x) & \text{Id} + \delta t\nabla_{t,d\perp}v_{t,d\perp}(x) \end{bmatrix}$. Now, to analyze the behavior of the scores, $s_t$, consider the change of variables formula for probability densities. Differentiating after taking the logarithm of the change of variables for probability densities, we have that $s_{t+1}(x_{t+1}) = s_t(x_t)(dF_t(x_t))^{-1} - \text{tr}(dF_t(x_t)^{-1}d^2F_t(x_t))(dF_t(x_t))^{-1}$. We now define

the evolution of the score components along $E_d^t$ : Using the covariance of $E_t^d$, we get,

$$(s_{t+1} \, E_{t+1}^d) \circ F_t = s_t \, E_t^d \, R_t^{-1} - \text{tr}((dF_t^{-1} d^2 F_t) \, E_t^d R_t^{-1}. \tag{3}$$

Since $E^d$ is covariant, we may interpret (3) as an operator acting on score component functions, $s_{t+1} E_{t+1}^d$, ($d$-dimensional row vectors at each $x$) at each time,

$$\mathcal{G}_t(p) \circ F_t = p \, R_t^{-1} - w_t \, E_t^d \, R_t^{-1}, \tag{4}$$

where, for convenience, we have defined, $w_t(x) = \text{tr}((dF_t(x)^{-1} d^2 F_t(x)) \in \mathbb{R}^D$. At the start, since $E_0^d$ is random, we may assume that $p$ is a zero vector field. Since the dynamics is compressive overall, $\min_{t \leq \tau} \alpha_t < 1$, and further, $\prod_t \alpha_t < 1$. However, at small $t$, the vector field $E_0^d$ is random, and by assumption, $v_t$ is uniformly contractive in all directions. Thus, $d^2 v_t$ has a small norm. Thus, we assume that $\|w_t\| \leq c \, \delta t$. Thus, in the starting phase we have, $\sum_{t \leq t^*} \|w_t E_t^d R_t^{-1} \cdots R_\tau^{-1}\| \leq c \, t^* \, \delta t \prod_{t \leq \tau} \alpha_t^{-1}$. Then, from (i)-(iii), $\|w_t E_t^d\| \leq \delta^{1/(\tau-t)} \, \delta t$, (see Appendix E) for any $t \geq t^*$, and so, $\|w_t E_t^d R_t^{-1} \cdots R_\tau^{-1}\| \leq \delta t \, c_1 \, \delta^{1/(\tau-t)} \, (1 + \delta)^{-\tau+t}$. Now, applying (4) recursively, we obtain that $\|\mathcal{G}^t(0) \circ F^t\| = \|\mathcal{G}^t(0)\| = \|\sum_{t \leq \tau} w_t \, E_t^d \, R_t^{-1} \, R_{t+1}^{-1} \cdots R_t^{-1}\| \leq c \, t^* \, \delta t \prod_{t \leq \tau} \alpha_t^{-1} + \delta t \, c_1 \sum_{t > t^*} \delta^{1/(\tau-t)} \, (1 + \delta)^{-\tau+t} = \mathcal{O}(\delta t)$. $\qquad \square$

**Implications for manifold learning.** The above theorem outlines sufficient conditions for the alignment of the most sensitive directions with the data manifold. There are two important consequences of this result. First, for an aligned and convergent generative model, we expect that any small learning errors in $v_t$ will result in probability mass redistributed on approximately the same support. In other words, the predicted density will have the approximately the same support as the target, since the directions orthogonal to the support are superstable (very large finite-time LEs). This is observed in Figure 3, where a generated digit (second column) is perturbed in the direction of the 1st LV (3rd column) to obtain another recognizable digit. On the other hand, when the 100th LV (shown in the 4th column of Figure 3) is added, we obtain artifacts that represent leaving the data manifold. In practice, the Lyapunov vectors may be computed using standard algorithms akin to iterative algorithms for eigenvectors (see [20, 7, 32]). Thus, an aligned generative model can effectively be used to compute the tangent bundle of the data manifold. In other words, an aligned generative model can learn the data manifold with the same sample complexity as the generative model, which is stronger than thought in previous works [54, 47], which have only shown that generative models learn the dimension of the manifold.

**Regularity of alignment.** Secondly, the alignment property itself is robust, i.e., when a generating process has the alignment property, under small perturbations, this property is retained, as we show next. This implies that, even an approximate generating process can be used for manifold learning.

**Lemma 4.4.** *The alignment property is regular.*

*Proof.* Let $F_0^\tau$ be a continuously differentiable generating process (i.e., $F_0^\tau \in C^1(M)$) for which alignment holds. Now, consider a sequence $\epsilon_k \to 0$, as $k \to \infty$, and a sequence of $C^1$ generating processes, $F_{\epsilon_k}^\tau$ that converge to $F_0^\tau$ in the $C^1$ norm. Let $E_0^d$ be differentiable (on some open set containing $M$ in $\mathbb{R}^D$). Starting with the same $E_0^d$, we may define the most sensitive subspaces, $E_{\tau, \epsilon_k}^d$ for each map $F_{\epsilon_k}^\tau$ as per the construction in this section. Then, from the continuity of $dF^\tau$, the covariance of $E_t^d$, and the continuity of $R^\tau$ (see Appendix F), each element of the sequence $E_{\tau, \epsilon_k}^d$ is locally continuous (we need to also assume the non-degeneracy of the stretching/compression factors, see Appendix F). Since $M$ is compact, from the Arzela-Ascoli theorem, we can conclude that $E_{\tau, \epsilon_k}^d$ contains a converging subsequence, which retains the alignment property. $\qquad \square$

**Justification for sufficient conditions.** We now describe the type of dynamics of the generating process that would satisfy the sufficient conditions in the theorem above. In the beginning, in the absence of information about the target score, the vector field $v_t$ could be uniformly compressive, e.g., in an SGM with the source density being nearly a Gaussian [47, 16]. Then, as information about the target is used, the score acquires an anisotropic behavior, and our sufficient conditions imply that correspondingly an anisotropicity must emerge in the vector field as well. Further, this anisotropicity in the vector field must be such that the vector field has small cross-derivatives. That is, while $\nabla_{t,d} v_{t,d}$ and $\nabla_{t,d\perp} v_{t,d\perp}$ can have large norms, the cross-derivatives, $\nabla_{t,d\perp} v_{t,d}$ and $\nabla_{t,d} v_{t,d\perp}$

must be negligible at the end. An SGM typically satisfies this condition, as one can visually see in Figure 1. In the leftmost image in Figure 1, which represents an intermediate time, the score vector field (which is equivalent to $v_t$ in an SGM) appears to stretch/compress differential volumes (i.e., it is anisotropic), while at the end (second image in Figure 1), there is very large compression toward the data manifold (the two moons) and the cross-derivatives (in local coordinates) are evidently small. The large compression toward the data manifold has been described from many perspectives before: [13, 26] use Fourier analysis, [47] uses stochastic analysis, [54, 41] take the statistical learning and uncertainty quantification perspective. Consistent with these results, our sufficient conditions (Theorem 4.3), when applied to SGMs, imply that the attraction/compression in volumes normal to the data manifold leads to robustness of the support. Notice that under this attraction condition, and (i)-(iii) of Theorem 4.3, we cannot control the norm of the orthogonal component of the score, $s_t E_t^{d\perp}$. This is due to two self-reinforcing effects: the compression factor matrices along $E_t^d$ have a smaller norm and hence a larger inverse compared to $R_t$. Secondly, the components of $w_t$ do not become small along $E_t^{d\perp}$ because the vector field may have non-negligible second derivatives, $\nabla_{t,d\perp}^2 v_{t,d\perp}$ (one can observe this visually for an SGM in the second column of Figure 1).

*Remark* 4.5 (Without the expansion assumption (i)). If $\min_t \alpha_t > 1$, notice that $\mathcal{G}_t \circ F_t$ in the proof above is a linear contraction in $\mathbb{R}^d$. In this case as well, the score component along $E_t^d$ decreases and becomes negligible. However, in practice, generating processes are compressive dynamics, as they acquire more information about the score when $t$ increases. Hence, we allow expansion for some part of the dynamics but assume compression overall. We need not assume any expansion at all, if an observed *phase transition* [1] occurs, wherein $F_t$ becomes linear for $t > t^*$, in which case, $w_t$ is the zero vector field.

# 5   Related Work

Tremendous progress has been achieved in the past few years to explain the empirical success of generative models. Several works, such as [15, 12, 34, 35, 10] have established theoretical guarantees for the convergence of diffusion models under different metrics (such as Wasserstein, reverse KL, total variation), including proving that they achieve minimax rates for learning the target [43], without demanding any functional inequalities (log-concavity) of the target. We do not tackle the question of generalization or convergence in this paper; instead, for small perturbations of generative models that indeed converge, we focus on the related but different question of when the predicted support is still close to the support of the target. Since our approach is dynamical, it applies to any other path on probability space that leads to the target. Hence, our analysis also applies to normalizing flows [45, 46] and flow matching variants [37, 56], which can be interpreted as easier-to-train models with specified probability paths.

Our sufficient conditions in Theorem 4.3 are consistent with observations made from several different angles on the generative process. For instance, [36] finds the emergence of linear behavior when the diffusion model starts to generalize, which is consistent with second derivatives of the vector field being small. Other works such as [8, 1, 62] study phase transitions in the dynamics or regularization effects [6] that leads to generalization.

Our work is most closely inspired by analyses and empirical evidence in [47, 13], which suggest the robustness of the support in score-based generative models in the context of the manifold hypothesis [48]. Here, we analyze the dynamics of the reverse process in a way that applies even to non-singular distributions. Further, our splitting in the Lyapunov directions suggests that there is more quantitative geometric information in the generating process about the data manifold beyond just the data dimension [54]. Although we do not focus on the unboundedness of the score (since our analysis applies also to targets with density) [47, 39], our results are consistent with unbounded score components along $E_\tau^{d\perp}$, normal to the data manifold (see section 4). We remark that finite-time Lyapunov analysis has been classically used for perturbation analysis of fluid flows in the geophysical fluids literature [50, 21, 33]. The eigenvectors of the Cauchy-Green deformation tensor, which in our notation is $dF^\tau \, dF^{\tau\top}$, are our Lyapunov vectors; in fluids, these have been used to understand the deformation in the current velocity field due to perturbations/strains in the past. Interestingly, the application of this analysis technique to generative modeling reveals insight into stability to errors in probability flow ODEs.

Finally, even though we do not explicitly discuss class-conditional generation and guided diffusions [60, 24], our work can potentially guide algorithms for learning projected scores or diffusions on

a lower-dimensional latent space [27, 9]. Our results establish a theoretical foundation for such projections of the vector field by uncovering the connection between the dynamics and the directions where accurate learning of the vector field is not necessary.

# 6 Numerical Results

Apart from the two-moons example we show in Figure 1, we collect many other two-dimensional examples that help with visualizing the vector field and the sufficient conditions in Theorem 4.3. In Appendix G.5, we give an example of a non-robust generating process. We observe that when using flow-matching [37, 56], with a source density being the 8 Gaussian density, we do not have the robustness property (see Appendix G.5 for more details, including the Lyapunov vector field). Among high-dimensional examples, we consider MNIST digit generation in Figure 3, but further examples are deferred to G. Using a pretrained model from a score-generative model repository [Github link], we find that adding perturbations of even large norm (when compared to the supremum norm over the pixel values) leads to high-quality images with comparable likelihood scores. Thus, consistent with observations in [47], we find that SGMs satisfy the robustness property. Finally, we describe an empirical observation that qualitatively confirms our manifold learning results from section 4. For an aligned generative model like the MNIST SGM (see Appendix G.3 for training details), we find that the Lyapunov exponents also provide geometric insight into the data manifold. A small gap arises in the LEs (Figure 3 Bottom) at an index consistent with previous estimates of the effective dimension of the MNIST data distribution [48]. Beyond index $\mathcal{O}(20)$, the LEs along the more stable directions appear to be continuous, while the top LEs are degenerate, depicting the deformations/perturbations of the most sensitive subspace of the underlying *feature space*.

# 7 Conclusion and limitations

Overall, we study the robustness of the support of the density predicted by a generative model, when the underlying vector field (score/drift) is learned with errors. Our results suggest that the tangent spaces of the support being aligned with the most sensitive Lyapunov subspaces leads to robustness (Proposition 4.1 and Theorem 4.3). Since the Lyapunov vectors are efficient to compute, aligned generative models can be used for manifold learning. The computation of LVs can also help us quantitatively distinguish between generative models based on their robustness property. Our proof techniques involve a novel combination of statistical learning with the finite-time perturbation theory of non-autonomous dynamical systems, which could be independently applicable in other settings. Here are some specific practical impacts of our results.

**Preserving invariances**: for singular targets in science, the support is often defined by some invariances (conservation laws) and group symmetries. Our results imply that ensuring alignment yields predicted targets that obey these known physical laws.

**Controlling GMs:** the most sensitive subspaces we introduce in this work can help design controllers for generative models to bias the target distribution for downstream applications, such as rare event sampling or unlearning. While we defer the details to future work, consider a vector field, $u_t$, representing a control neural network. Given an orbit, say, $x_t$, that leads to an unwanted sample at time $\tau$, our goal will be to train $u_t$ such that $x_t + u_t(x_t)$ is a desired orbit. To do this in reduced dimensions, we only need to determine the components $u_t E_t^d$ in the most sensitive directions.

The wider implication of our results is that the theory of dynamical systems (including perturbation theory and the operator theoretic view) can advance our understanding of as well as provide principled algorithmic improvements to generative modeling and more broadly of probability flow dynamics.

**Limitations.** We only provide a sufficient condition and not a necessary condition for alignment and hence robustness. More extensive experimentation with various different generative models is needed to determine the most common scenario where alignment occurs. More advanced and extensive experiments are also needed to understand the prevalence of alignment and therefore robustness in practice. Further, our method of detecting tangent spaces of the data manifold hinges on there being alignment and our results do not explicitly reveal insight into memorization.

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

# A  Background on the random dynamical systems view of diffusion models

In the main text, we define the generating process of any generative modeling algorithm as a random dynamical system. In particular, dynamical generative models like normalizing flow, rectified flow, conditional flow matching-variants, stochastic interpolants [46, 37, 56, 38] etc can be viewed as nonautonomous (forced in a time-dependent manner) and deterministic dynamical system. On the other hand, diffusion models or score-based generative models [18, 60, 10, 42, 52] have stochastic generating processes (reverse or denoising process). However, in the main text, we argued that, for the purpose of analyzing the probability flow, we may ignore the noise in the reverse process, and thus, also consider SGMs to be nonautonomous but deterministic systems. Here, we present a more general dynamical systems definition applicable to both deterministic and stochastic nonautnomous systems. These, so-called random dynamical systems have been classically studied as part of dynamical systems theory, while undergoing parallel development in the probability and stochastic analysis communities (see e.g., [5] and [31] for textbook expositions of random dynamical systems from the dynamical systems/ergodic theory and probabilistic/stochastic analysis perspectives respectively).

The unifying random dynamical systems framework to represent generative models is as follows. Consider an instance of a time-discretized Wiener path, $\Xi = \{\xi_0, \cdots, \xi_{T-1}\}$, where $\xi_i$ are independent standard normal variables. These provide stochastic forcing to the dynamics , $F_t^\Xi$, at time $t$, which is now extended with a superscript $\Xi$ to indicate a fixed noise path. For a fixed noise, $\Xi$, $F^{\tau,\Xi}$, is defined as a composition (time $\tau$-dynamics) as expected, $F^{\tau,\Xi} = F_{\tau-1}^\Xi \circ F^{\tau-1,\Xi}$, with $F^{0,\Xi} = \text{Id}$, for all $\Xi$.

**Example: score-based diffusion [51, 52]** In score-based diffusion models, the generating process is an Ito process of the form: $dX_t = f_t(X_t)\ dt + dW_t$, where $f_t$ is a deterministic score term that is represented as a neural network, and $W_t$ is a Wiener path/Brownian noise. Given a time-integration scheme for this stochastic process, we can define $F_{t,\Xi}$ as the stochastic flow over a short time. For instance, using Euler-Maruyama time-integration with a fixed timestep, $\delta t$, we have $X_{t+1} = X_t + f_t(X_t)\ \delta t + \sqrt{\delta t}\ \xi_t$. Then, $F_t^\Xi(x) := x + \delta t f_t(x) + \sqrt{\delta t}\ \xi_t$. In summary, we view a stochastic generating process as a one-parameter family of random diffeomorphisms, $F_t^\Xi$, for (almost) every time sampling, $\Xi$, of the underlying Brownian path. For the existence of this one-parameter family, we refer to classical works on stochastic flows [31, 30]. With this RDS view, the stochastic process (a continuous-state discrete-time Markov chain) has a time-dependent transition kernel that can now be written in terms of $F^\Xi$ as:

$$P_t(X_{t+1} \in A | X_t = x) = \mathbb{P}(\xi : F_t^\Xi(x) \in A).$$

Substituting for $F_t^\Xi(x) = x + \delta t f_t(x) + \sqrt{\delta t}\ \xi_t$, and using the fact that $\xi_t$ has a normal distribution, we get, $P_t(A|x) = \int_A e^{-\|y-x-\delta t f_t(x)\|^2/(2\delta t)}\ dy$ for this example process. The usual equation for the evolution of the probability measures, say $\mu_t$, is the Kolmogorov forward equation, which is given by,

$$\mu_{t+1}(A) = \int P_t(A|x) d\mu_t(x) = \int \mathbb{P}(\xi_t : F_t^\Xi(x) \in A)\ d\mu_t(x). \tag{5}$$

On the right hand side of the above equation, notice the transition kernels written in terms of the RDS. Moreover, beyond $\mu_t$, we can also define a sequence of *sample-path measures*, $\mu_t^\Xi$, which are obtained for fixed Brownian paths via pushforwards or the *Frobenius-Perron* operator,

$$\mu_{t+1}^\Xi = F_{t\sharp}^\Xi \mu_t^\Xi := \mu_t^\Xi \circ F_t^{\Xi,-1}. \tag{6}$$

Since we start the process with $X_0 \sim \mu_0$, we take $\mu_0^\Xi = \mu_0$ for all paths $\Xi$. We note that since $\mu_0$ typically has a density (with respect to Lebesgue $\mathbb{R}^d$), say, $\rho_0$, $\mu_t$ as well as the sample-path measures, $\mu_t^\Xi$ also have densities up to a finite time, even when $F_t^\Xi$ is a non-volume preserving diffeomorphism. We denote these densities as $\rho_t$ and $\rho_t^\Xi$ respectively corresponding to $\mu_t$ and $\mu_t^\Xi$. With the density $\rho_t^\Xi$ defined, we can use the change-of-variables formula in (6) to obtain,

$$\rho_{t+1}^\Xi = \mathcal{L}_t^\Xi \rho_t^\Xi := \frac{\rho_t^\Xi \circ F_t^{\Xi,-1}}{|\det \nabla F_t^\Xi| \circ F_t^{\Xi,-1}}, \tag{7}$$

where we define $\mathcal{L}_t^\Xi$ to be a time-dependent linear operator that transforms densities through a deterministic system. Combining with the Kolmogorov forward equation in (5), we also have,

$$\rho_{t+1}(y) = \mathbb{E}_\Xi \mathcal{L}_t^\Xi \rho_t^\Xi, \tag{8}$$

provided $\rho_0 = \rho_0^{\bar{\Xi}}$, where we have used $\mathbb{E}_\Xi$ to denote expectation with respect to the independent standard Gaussian RVs, $\Xi = [\xi_0, \cdots, \xi_{\tau-1}]$.

For a fixed noise $\Xi$, the deterministic dynamics $F^{\tau,\Xi}$ is a coupling between $\rho_0$ and a noise path-dependent density $\rho_\tau^\Xi$, i.e., $\mathcal{L}^{\tau,\Xi}\rho_0 = \rho_\tau^\Xi$. Here, the operator $\mathcal{L}^{\tau,\Xi}$ is called the *Frobenius-Perron* or transfer operator, which describes the evolution of probability densities through the map, $F^{\tau,\Xi}$. The Frobenius-Perron operator $\mathcal{L}^{\tau,\Xi}$ is also defined as a composition of per-iteration operators, which we denote by $\mathcal{L}_t^\Xi$, so that $\mathcal{L}_t^\Xi \rho_t^\Xi = \rho_{t+1}^\Xi$. Specifically, we define $\mathcal{L}_t^\Xi \rho = (\rho\,\Delta\mathrm{vol}_t) \circ F_t^{\Xi^{-1}}$, where $\Delta\mathrm{vol}_t(x) = |\det(dF_t^\Xi)|^{-1}$ indicates the change of differential volume under the application of the map $F_t^\Xi$. Note that, since the noise $\Xi$ is independent of the state, $\Delta\mathrm{vol}_t$ is not a function of $\Xi$. In the standard stochastic analysis literature, we generally refer to $\mathbb{E}_\Xi \mathcal{L}_t^\Xi$ as the *Kolmogorov forward* operator, which is described in (5) when $\mu_t$ are absolutely continuous with respect to Lebesgue. By definition, $\mathbb{E}_\Xi \mathcal{L}_t^\Xi \rho_t = \rho_{t+1}$. Classically, we may write, $\rho_{t+1}(x) = \int_M \kappa_t(x, y)\, \rho_t(y)\, dy$, where $\kappa_t(x, y)$ is the conditional density of the transition kernel, $P_t(y, dx)$, which represents the conditional density of the state at time $t + 1$ being $x$ conditioned on the state at time $t$ being $y$. This assumes the kernel is absolutely continuous in both arguments, which is typical for diffusion-based models (e.g., the transition probability measure in (5) is absolutely continuous). When the target measure, $\mu_\tau = p_{\mathrm{data}}$ is singular, making $\rho_\tau$ undefined, the probability density $\rho_{\tau-\Delta}$ for a small $\Delta$ approximates a notion of density associated with the target. For simplicity, $\rho_\tau$ in this case should be interpreted as $\rho_{\tau-\Delta}$, which is a convolution of the target measure, $p_{\mathrm{data}}$, with a Gaussian of variance $\Delta$.

## A.1 Diffusion models

Our paper treats the reverse process of a diffusion model as a random dynamical system. While we presented this view in the main text and the previous section, here we review the more standard view through SDEs. Diffusion models generate samples from an unknown *target* probability distribution $\pi \in \mathcal{P}(\mathbb{R}^D)$ from which we only have access to samples. The general setup [52] is to consider a diffusion process, which will be referred to as the *forward process*, that transforms the target into a distribution that is easy to sample from. Typically, the forward process is chosen from a class of Ornstein-Uhlenbeck processes

$$dX_t = -\beta_t X_t\, dt + \sqrt{2\beta_t} dB, \quad X_0 \sim \pi. \tag{9}$$

It is assumed that $\beta_t$ is positive and integrable such that the integral $\int_0^t \beta_s\, ds \to \infty$ as $t \to \infty$. It follows that (9) is a time-rescaling of the standard Ornstein-Uhlenbeck process, through the time change of variables $\tau = \int_0^t \beta_s\, ds$ and the marginals $\rho_t$ converge geometrically to the standard multivariate normal distribution $N(0, I_D) \in \mathcal{P}(\mathbb{R}^D)$. Since (9) has linear drift, it follows that the solutions can be solved analytically, yielding the formula for the marginals in terms of the target $\rho_t(x) = \mathbb{E}_{X_0 \sim \pi}[\rho_t(x|X_0)]$ with the conditional density given by the kernel

$$\rho_t(\cdot|x_0) = N\left(\exp\left(-\int_0^t \beta_s\, ds\right) x_0;\ \left(1 - \exp\left(-2\int_0^t \beta_s\, ds\right)\right) I_D\right). \tag{10}$$

We note here that the above kernel is smooth in the space variable, implying the $C^\infty$ smoothness for the marginals for all $t > 0$.

The forward process is ergodic, with the marginals converging to the standard normal at rate $\exp\left(-\int_0^t \beta_s\, ds\right)$. After a finite large time $T$ samples are assumed to be approximately normal. Once $T$ is chosen, define the time-reversed process $Y_t := X_{T-t}, t \in [0, T)$. It is known ([22], [4]) that $Y_t$ is a Markov process and that it is a weak solution to the following stochastic integral

$$dY_t = \beta_{T-t}\left(Y_t + 2\nabla \log \rho_{T-t}(Y_t)\right) dt + \sqrt{2\beta_{T-t}} dB_t, \quad Y_0 \sim \rho_T. \tag{11}$$

From the smoothness of the marginals $\rho_t$ — and hence smoothness of the drift term $y + 2\nabla \log \rho_t(y)$ — it follows that (11) admits a strong solution for times $t < T$. It follows from weak uniqueness of the backward process [44] that the law of $Y_{T-t}$ coincides with that of $X_t$. Hence, generating trajectories from the reverse process provides a way of sampling from the target distribution as $t \to T$.

The backward process is only defined for times $t < T$. In order to extend the backward process to the full time interval $t \in [0, T]$, one needs the assumption on the initial density that $\nabla \log \rho_0 = \nabla \log \pi$

exists in a weak $L^2$ sense [22]. However, in practice this is almost never satisfied as the target density is typically singular. This implies a singularity in the score $s_t$ that grows as $\frac{1}{\int_0^t \beta_s\, ds}$ as $t \to 0$. To overcome this issue, the backward process is typically only sampled up to time $t = T - \Delta$, which is responsible for the characteristic noise typically present in image models.

## B    Response of the predicted density to learning errors

In the main paper, we argued that we may ignore the noise term $\xi$ for a fixed path in probability space, and simply consider the deterministic nonautonomous system. Here we show how to extend the perturbation response result in section 3 to random dynamical systems. Using the framework presented in section A, we may go through the same derivation as in section 3 pathwise, by replacing $\mathcal{L}_t$ with $\mathcal{L}_t^\Xi$. Again, the density $\rho_\tau^\Xi$, is close to the target (on averaging with respect to the noise paths, $\Xi$), but not exactly equal. In case the target density with respect to Lebesgue does not exist, we can perform integration by parts and treat $\rho_\tau^\Xi$ as a genuine density due to the convolution of the target measure with Gaussians that describes the $\rho_\tau^\Xi$ in the discrete time algorithm (DDPM).

As before, we consider $f$ to be constant functions on the data manifold that are differentiably extended to $\mathbb{R}^d$. The pathwise responses derived in this way contain pathwise score functions, $s^\Xi$ which are not the same as the score functions, $s$. While $\mathbb{E}\rho^\Xi = \rho$, we do not get the score by taking expectations of the pathwise scores. In order to compute $s^\Xi$ however, we may recursively apply the log gradient of the change of variables through the map, $F^\Xi$, i.e., $\mathcal{L}_t^\Xi$. The above pathwise statistical response, if uniformly bounded over the Wiener paths, due to dominated convergence, allows us to exchange limits, and thus, the overall statistical response can still be computed pathwise via,

$$\langle f, \partial_\epsilon|_{\epsilon=0}\mathbb{E}_\Xi\, \mathcal{L}_\epsilon^{T,\Xi}\rho_0\rangle = \langle f, \mathbb{E}_\Xi\, \partial_\epsilon\mathcal{L}_\epsilon^{T,\Xi}|_{\epsilon=0}\rho_0\rangle. \tag{12}$$

## C    Tangent dynamics: evolution of infinitesimal perturbations

The primary objective of this work is to study the effect of learning errors on the dynamics. For stochastic generative processes, we can extend the linear perturbation analysis in the main text to each noise realization of an RDS. As before, to model the effect of score learning errors, we consider evolving $F_{t,\Xi}$ with perturbed scores of the form, $s_t + \epsilon\chi_t$, where $\chi_t$ is a time-dependent vector field that indicates the direction of the error in the score. The perturbed dynamics, for a fixed noise path, is represented as, $F_\epsilon^{t,\Xi} = F_{t-1,\epsilon}^\Xi \circ \cdots \circ \cdots F_0^\Xi$, and correspondingly, the perturbed densities, by $F_{\epsilon\sharp}^{t,\Xi}\rho_0 = \rho_{t,\epsilon}^\Xi$, leading to the perturbed predicted density, $\rho_{\tau,\epsilon}$, when we take an expectation over noise realizations $\Xi$. We can set $\zeta_t := \partial_\epsilon F_\epsilon^{t,\Xi}$ to represent a time-dependent vector field that gives the perturbation in the state (sample) at time $t$ due to the learning error field. Taking $\epsilon \to 0$, we can obtain the following recursive relationship for $\zeta_t$ :

$$\zeta_{t+1} \circ F_t^\Xi = dF_t^\Xi\, \zeta_t + \chi_t \circ F_t^\Xi, \tag{13}$$

simply by applying chain rule. Unrolling this recursion, and since $\zeta_0^\Xi = \partial_\epsilon F_\epsilon^{0,\Xi} = \partial_\epsilon\mathrm{Id} = 0$ identically as a vector field, we obtain,

$$\zeta_{t+1}^\Xi \circ F_t = \sum_{n=0}^t dF_t\, dF_{t-1} \circ F_{t-1}^{-1}\cdots dF_{n+1} \circ F_{n+1}^{-1} \circ \cdots \circ F_{t-1}^{-1}\, \chi_n. \tag{14}$$

A vector field can be evaluated at a specific point, say $x \in \mathbb{R}^D$, to give a *tangent vector*, that indicates the direction of infinitesimal change at $x$. An interpretation of this infinitesimal change when viewed through differentiable scalar fields is the following. If $g : \mathbb{R}^D \to \mathbb{R}$ is a scalar function on the domain, then, at $x$, a vector field represents one among the possible directions of an infinitesimal change in $g$. In other words, tangent vector fields can be thought of as (linear) operators which when acting on differentiable functions produce their directional derivatives at each point. As an example, $\zeta_t(x) \in \mathbb{R}^D$ is a tangent vector that can be used to produce the directional derivative of any $g$, as $dg(x) \cdot \zeta_t(x) := \lim_{\epsilon\to 0}(g(x + \epsilon\zeta_t(x)) - g(x))/\epsilon$. In this sense, there is a natural interpretation for the sequence of vector fields defined in (13). Let us fix an orbit/sample path, $\{x_t = F_t^\Xi(x_{t-1})\}$. The tangent vectors $\zeta_t(x_t) \in \mathbb{R}^D$ can be applied to a scalar function $g$ to obtain the overall infinitesimal change in $g$ along the sample path due to infinitesimal learning errors. More precisely,

$$\partial_\epsilon(g \circ F^{t,\Xi})(x_0) = dg(x_t) \cdot \zeta_t(x_t). \tag{15}$$

Rewriting (14) to make $\zeta_t(x_t)$ explicit along a fixed sample path,

$$\zeta_t(x_t) = \sum_{n=0}^{t-1} dF_{t-1}(x_{t-1}) \cdots dF_{n+1}(x_{n+1}) \, \chi_n(x_{n+1}). \tag{16}$$

Each term in the above sum consists of multiplication by a sequence of matrices. Let us define $A_t := dF_t(x_t) \in \mathbb{R}^{D \times D}$ and the product $A_{n,t} := A_t A_{t-1} \cdots A_n$, for $0 \leq n \leq t-1$, for the sake of shorter notation. That is, the perturbation vector $\zeta_t(x_t)$ can now be written as

$$\zeta_{t+1}(x_{t+1}) := \sum_{n=0}^{t} A_{n+1,t} \, \chi_n(x_{n+1}). \tag{17}$$

To analyze the effect of infinitesimal errors on infinitely long sample paths, we can let $n \to -\infty$ in the above equation. In this case, the asymptotic behavior of the product of random matrices comes into play. Oseledets theory (see e.g., [5]) is a collection of classical results on random matrix products as applied to cocycles defined on dynamical systems. Essentially, assuming that $\max\{0, \log \|A_t\|\}$ is summable for almost all paths, one can define Lyapunov exponents (for each $\Xi$) to be the logarithms of the set of eigenvalues of the matrix, $\lim_{n \to -\infty} (A_{n,t}^\top A_{n,t})^{1/2(t-n)}$. Corresponding to the Lyapunov exponents (LE), there is a decomposition of the tangent space at each $t$ in the characteristic directions called Oseledets subspaces, i.e., directions in which the perturbation norms grow at an exponential rate corresponding to a given LE. Thus, to analyze the growth/decay of the norms in the time-dependent linear dynamical system (13), these characteristic directions form a natural basis. Here, since our dynamical system is defined only over a finite time interval, we consider a computational proxy for the Oseledets spaces, which are described in the main text (section 4). In the remainder of this section, we let $n \to -\infty$ and review Oseledets theorem.

Ignoring the control or forcing (inhomogeneous) term in 13, to isolate the time-asymptotic growth/decay on an exponential scale, we can consider the following homogeneous tangent equation,

$$\omega_{t+1} = A_t \, \omega_t. \tag{18}$$

If we are only interested in growth/decay on an exponential (in $t$) scale, finite sums for $n$ close to $t$ in (17) are not significant. Moreover, the vectors $\chi_n(x_{n+1})$ are path-dependent and perturbation-dependent, and they are not fundamental directions characteristic of the dynamics. Thus, by considering a decomposition (as in section 4) of $\chi_t(x_t)$ along Oseledets spaces, we can provide a pessimistic analysis, since a random vector $\chi_t(x_t)$ will, with probability 1, have a non-zero component in the leading Oseledets space at $x_t$.

The homogeneous tangent equation (18) gives the evolution of infinitesimal perturbations in the initial conditions, i.e., $\omega_t := dF^{t,\Xi} \, \omega_0$.. This equation gives the most general evolution of infinitesimal perturbations along a generic sample path $\{x_t\}$. When $x_t$ is invariant, i.e., a fixed point, $A_t$ is also invariant, and this reduces to linear stability analysis. When $x_t$ is a periodic orbit, the matrices $A_t$ are classically studied with Floquet theory and corresponding exponents. In more generality, the random matrix product $A_{n,t}(x_n) : T_{x_n}\mathbb{R}^D \to T_{x_t}\mathbb{R}^D$, known as the *tangent propagator* [32], is studied as $n \to -\infty$ under Oseledets multiplicative ergodic theorem.

When the dynamics $F_t$ is invertible, we consider the limit

$$W^-(t) = \lim_{n \to -\infty} \left( A_{n,t}^{-\top} A_{n,t}^{-1} \right)^{1/(2(t-n))}.$$

The eigenvectors $\phi_i(t)$ of $W^-(t)$ are called the *backward Lyapunov vectors (BLVs)*, and the negative log of the singular values $\lambda_i = -\log \sigma_i$ are called the Lyapunov exponents. Conventionally, the LEs are still deterministic and are defined by taking expectations with respect to the noise paths $\Xi$. The vectors $\phi_i(t)$ form a basis for the tangent space at $x_t$ are defined for $\mathbb{P}$-a.e. (for almost every noise realization). For an exposition on the ergodic theory for RDS, see [29]. When the distribution $\mathbb{P}$ does not depend on time, the Backward Lyapunov vectors can also be defined in a deterministic manner $\mathbb{P} - a.e.$

## D  Robustness of the support upon alignment

Proposition 4.1 shows that with high probability an aligned and convergent generative model can be used to learn the support of the data distribution accurately. First, by convergence, we mean that the

generating process enjoys a theoretical convergence result in Wasserstein metric. For instance, we can consider a convergence result from [34] for denoising diffusion probabilistic model (DDPM), a time-discrete diffusion model. For any general target with compact support, as we have assumed, suppose the score is learned with a $L^2$ error $\mathcal{O}(\epsilon)$, uniformly over time $t \leq \tau$. Then, [34] show that the Wasserstein-2 distance between the predicted density, $\rho_{\tau,\epsilon}$ and the target $p_{\text{data}}$ is $\mathcal{O}(\epsilon^{1/18})$. Note that, by definition of Wasserstein-2 distance, if $T$ is the optimal transport map between $p_{\text{data}}$ and $\rho_{\tau,\epsilon}$, then, $E_{x \sim p_{\text{data}}} \|T(x) - x\|^2 \leq C\epsilon'$, where $T(x) \sim \rho_{\tau,\epsilon}$. Now since $\|T(x) - x\|$ is a random variable with a small mean and variance, we can get an $\epsilon_0$ (applying Chebyshev's inequality e.g.) in the statement of Proposition 4.1 given any $\delta > 0$, such that with probability (over $n$ independent draws from $p_{\text{data}}$) $\geq 1 - \delta/2$, we have that $\|T(x_i) - x_i\| \leq \epsilon_0$, for all $i \leq n$.

Next we examine the alignment property. In Proposition 4.1, we assume alignment with high probability. That is, with probability $\geq 1 - \delta/2$ over independent draws from $p_{\text{data}}$, alignment holds, i.e., at the generated samples, $T(x_i)$, the most sensitive Lyapunov subspace $E^d$ is tangent to the support of $p_{\text{data}}$. In other words, the generated samples $T(x_i) = x_i + \epsilon h_i$, where $h_i$ is along $T \partial M$. Now consider a one-classifier trained to predict 1 if a data point is on the support and -1 otherwise. A kernel-based classifier is always realizable for a discrete data distribution [49]. It is a one-class classifier because for all the data points $x_i$, the output label is 1 and we do not have negative samples.

A key observation is that the confidence margin of a (one-class) hyperplane classifier trained using $x_i$ is the same as that trained using $T(x_i)$. Therefore, we can apply a known generalization result, and going further, even data-dependent upper and lower bounds for classification using the true data distribution to now the predicted distribution, provided the prediction is aligned (margin does not change). This is the essence of the proof. In summary, we pose learning the support as estimating a one-class classifier. Then, we use the fact that the margin does not change when we move data points along the separating hyperplane.

# E   Alignment proofs

In the proof of Theorem 4.3, we make assumptions about the dynamics of the vector field $v_t$, whose time-discretized flow is our dynamics, $F^t$. Mainly, toward the end, when $t > t^*$, we assume specific anisotropic behavior of the vector field. It is helpful to think of the anisotropy by considering local coordinates that align the first $d$ coordinates with the most sensitive subspaces, $E_t^d$. In other words, consider local coordinates, $\Phi_t : \mathbb{R}^D \to \mathbb{R}^D$ around $x_t$, such that, $\Phi_t(0) = x_t$ and $d\Phi_t(0)$ maps the first $d$ standard basis vectors to $E_t^d$.

Recall assumptions (i)-(iii) in the statement of Theorem 4.3. Consider the Jacobian matrix at time $t$, $dF_t(x)$ in block form, $dF_t(x) = \begin{bmatrix} \text{Id} + \delta t \, \nabla_{t,d} v_{t,d}(x) & \delta t \nabla_{t,d\perp} v_{t,d}(x) \\ \delta t \, \nabla_{t,d} v_{t,d\perp}(x) & \text{Id} + \delta t \nabla_{t,d\perp} v_{t,d\perp}(x) \end{bmatrix}$, and the second derivative $d^2 F_t$ can be written as two block tensors: $\begin{bmatrix} \delta t \, \nabla^2_{t,dd} v_{t,d}(x) & \delta t \, \nabla^2_{t,dd\perp} v_{t,d}(x) \\ \delta t \, \nabla^2_{t,dd} v_{t,d\perp}(x) & \delta t \, \nabla^2_{t,dd\perp} v_{t,d\perp}(x) \end{bmatrix}$ and $\begin{bmatrix} \delta t \, \nabla^2_{t,dd\perp} v_{t,d}(x) & \delta t \, \nabla^2_{t,d\perp d\perp} v_{t,d}(x) \\ \delta t \, \nabla^2_{t,dd\perp} v_{t,d\perp}(x) & \delta t \, \nabla^2_{t,dd\perp} v_{t,d\perp}(x) \end{bmatrix}$. To obtain an estimate of $w_t := \text{tr}((dF_t)^{-1} \, d^2 F_t)$, we first observe that using assumptions (ii)-(iii), the Schur complement of the first $d \times d$ block of $dF_t$ reduces to $\text{Id} + \delta t \, \nabla_{t,d\perp} v_{t,d\perp}$. Using this Schur complement and assumption iii, we obtain that the first block of $w_t$, which is $w_t E_t^d$ is given by $\delta t \, \text{tr}((\text{Id} + \delta t \, \nabla_{t,d} v_{t,d})^{-1} \, \nabla^2_{t,dd} v_{t,d})$. Then, using assumption ii, we obtain the estimate in the main text.

# F   Regularity of alignment

Lemma 4.4 shows a notion of regularity of the alignment property. We show this by using the Arzela-Ascoli theorem on the space of functions $E_\epsilon^d$, for some $\epsilon$ perturbation of the dynamics. Applying Arzela-Ascoli gives the existence of a converging subsequence on this space. This subsequence consists of most sensitive subspaces of perturbed systems, which from convergence, will also be closely aligned with the data manifold if the original dynamics is aligned. To apply Arzela-Ascoli, one sufficient condition is to assume $F_{t,\epsilon} \in C^{1+\alpha}$ since we then obtain that $E_\epsilon^d$ is Holder continuous. This is because $E_\epsilon^d$ is by construction an orthonormal basis for the column space of $dF_\epsilon^t$, which is

$C^\alpha$. For the Holder continuity of $E^d_\epsilon$, we also need the eigenvalues of $dF^t_\epsilon$ to be nondegenerate. With uniform Holder constants and exponents, since $M$ is compact, we get the needed equicontinuity.

## G  Additional numerical experiments

Our numerical results using score-based diffusions indicate robustness of support in all cases; further, they also show alignment, qualitatively validating the dynamical mechanism for robustness that we show in the main text. We report on the numerical methods, implementation details and our experiments in this section. The supplementary material also contains the code needed to reproduce the figures in the main text.

### G.1  Sampling via reverse diffusion

In the case of score-based diffusions, our dynamics $F^\tau$ refers to the Euler-Maruyama discretization of the reverse diffusion (11). There are various noise schedules $\beta_t$ used in practice. In terms of the continuous time SDE (9), choosing $\beta_t$ is tantamount to reparameterizing the time variable in the standard Ornstein-Uhlenbeck process via $\tau = \int_0^t \beta_s \, ds$. From a mathematical perspective, the density evolutions are therefore the same. Practically, however, the process has to be discretized and some noise schedules are more robust against time-discretization errors [23]. For the purpose of this study, we therefore fix the noise schedule to be the cosine noise schedule from [42] that was shown empirically to yield good FID and NLL scores. We observe that our experimental results on alignment and robustness do not change when using different noise schedules. The cosine noise schedule from [42] translates to the formula for $\beta_t$ given by

$$\beta_t = \frac{\pi}{(1+\delta)} \cdot \frac{\sin\left(\frac{\pi}{2} \cdot \frac{t+\delta}{1+\delta}\right)}{\cos\left(\frac{\pi}{2} \cdot \frac{t+\delta}{1+\delta}\right)}.$$

This comes from the formula for $\overline{\alpha}_t = f(t)/f(0)$, $f(t) = \cos\left(\frac{t+\delta}{1+\delta} \cdot \frac{\pi}{2}\right)^2$ given in [42] and noting that $\overline{\alpha}_t = \exp\left(-\int_0^t \beta_s \, ds\right)$.

Once a suitable approximation to the score is acquired, the backward equation (11) is discretized to yield the random dynamical system

$$Y_{n+1} = F_n(Y_n, \xi_n) := Y_n + \beta_{T-t_n}\left(Y_n + 2s_{T-t_n}(Y_n)\right)\delta t + \xi_n\sqrt{2\beta_{T-t_n}\delta t}, \quad Y_0 \sim N(0,1).$$

We also study solutions the *perturbed* system

$$Y_{n+1} = F_n(Y_n, \xi_n) := Y_n + \beta_{T-t_n}\left(Y_n + 2s_{T-t_n}(Y_n) + \epsilon\chi_{T-t_n}(Y_n)\right)\delta t + \xi_n\sqrt{2\beta_{T-t_n}\delta t}.$$

The perturbation vector $\chi_n$ specifying the error between the original dynamical system $F_n(\cdot, \xi_n) = F_n(\cdot, \xi_n; 0)$ and the perturbed dynamical system $F_n(\cdot, \xi_n; \epsilon)$, and $\epsilon$ measures the strength of the perturbation (see A). The timesteps $t_n$ is chosen equispaced with $0 < t_0 < \cdots t_n = T - \Delta = 1 - \Delta$, with $\Delta$ controlling the early stopping time to avoid singularities. This corresponds to solving the backward SDE (11) from $t = T - t_0$ backward to $t = \Delta$.

### G.2  Two dimensional examples

We perform a number of experiments on two-dimensional domains with one or two-dimensional support of the target. We show our experiments with the 2 moons distribution in Figures 1(main paper), 8 and 7. We also visualize the Lyapunov vectors on a different example in Figures 4 and 2. Throughout, LVs and LEs are computed using the QR algorithm (a finite-time version of the Gram-Schmidt process from [20]) described in section 4.

In these planar experiments, we represent the manifold as a curve (or a collection of curves as in the half-moon example) $\Gamma = \{\Gamma(t) : t \in [0,1]\} \subset \mathbb{R}^2$. The target measure is given by $dp_{\text{data}} = q \, d\gamma$ where $d\gamma$ is the arc-length measure for the curve $\Gamma(t)$, and $q$ some smooth density. We can compute expectations against $\pi$ via the parameterization as

$$\mathbb{E}[g(X)] = \int_\Gamma g(x)p(x)ds = \int_0^1 g(\Gamma(t))p(\Gamma(t))\Gamma'(t)\, dt.$$

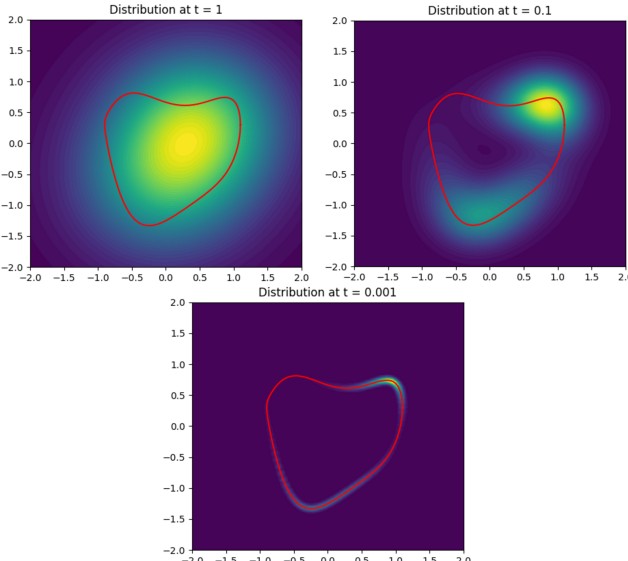

Figure 4: Score-based diffusion with numerical estimates of the score. Top row: starting density, $\rho_0$ and the density at time 0.9. Bottom row: predicted target, $\rho_{1-\Delta}$. In each figure, the red curve represents the analytical data manifold.

The Ornstein-Uhlenbeck process 9 is a linear SDE with additive noise. The density $\rho_t$ can therefore be solved analytically [44] via the integral equation

$$\rho_t(x) = \int_\Gamma \rho_t(x|x_0)q(x_0)ds.$$

The kernel $\rho_t(x|x_0)$ is the Green's function to the associated Fokker-Planck equation and is given by

$$\rho_t(x|x_0) = \frac{1}{Z_t}\exp\left(-\frac{|x - e^{-\frac{t}{2}}x_0|^2}{2(1-e^{-t})}\right).$$

The score $s_t = \nabla \log \rho_t$ can also be solved for in terms of the one dimensional integral

$$s_t(x) = \frac{\int_\Gamma \nabla_x \rho_t(x|x_0)q(x_0)ds}{\int_\Gamma \rho_t(x|x_0)q(x_0)ds}.$$

Our paper focuses on the propagation of score errors through the dynamics. To validate our theoretical results on the robustness of the support in a stylized setting, and since the integrals involved are tractable in the low-dimensional setting, we estimate the score via quadrature rather than training a neural network. This is done to maintain explicit control of the errors involved in our motivating examples and experiments.

### G.3  MNIST training details

Here we present additional details on the MNIST results from the main paper. We showed that MNIST generation with diffusion models tends to have robustness of the support. Further, we also observed that our proposed mechanism of alignment holds even in this higher dimensional setting. Specifically, we showed that the leading $\mathcal{O}(20)$ (approximately the intrinsic dimension of the support/data manifold) LVs span the tangent spaces to the data manifold. As empirical proof of this, we saw that moving along an LV of a higher index (indices are, by convention, in decreasing order of LEs) takes us off the data manifold. This is shown in more detail in Figure 5.

These images are produced by a DDPM where the score model is trained by minimizing the simplified conditioned score matching loss from [25]:

$$\mathcal{L}_{\text{simple}}(\theta) = \mathbb{E}_{t,x_0,\epsilon}\left[\|\epsilon - \epsilon_\theta(\sqrt{\alpha_t}x_0 + \sqrt{1-\alpha_t}\epsilon, t)\|^2\right],$$

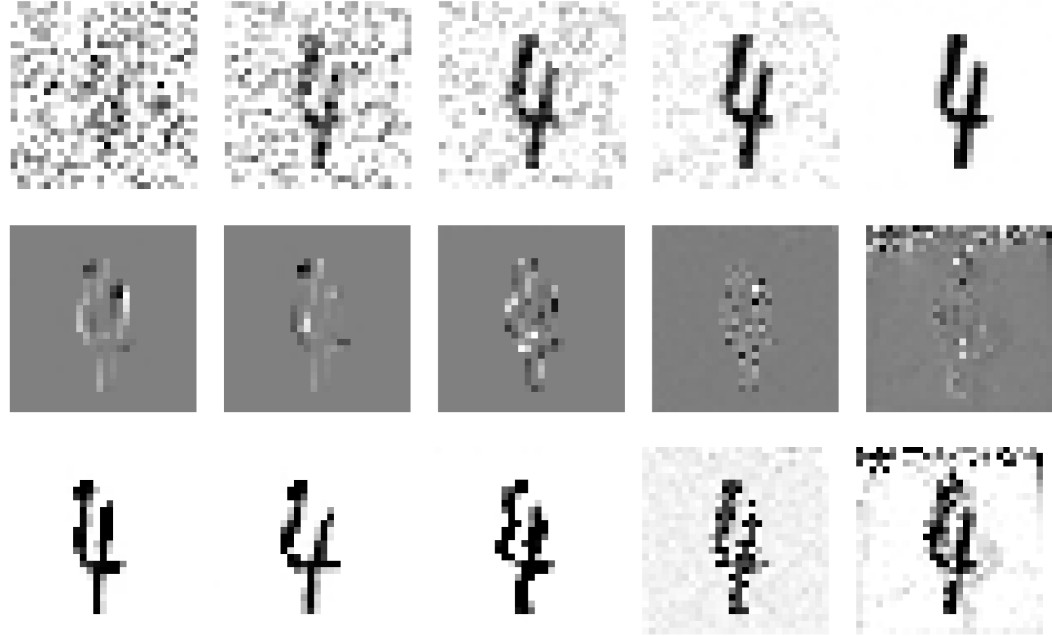

Figure 5: Top: Denoising diffusion trajectory sampled by approximating the score using a U-Net architecture trained on the MNIST digit dataset. Middle: The Lyapunov vectors of indices 1, 2, 20, 50 and 100 (from left to right) calculated along the sample trajectory. Notice that the principal Lyapunov vectors recover meaningful features of the sampled digit. The is in contrast to the lower Lyapunov vectors (higher indices indicate smaller LEs), which become progressively more noisy. Bottom: The sample image perturbed in the direction of the Lyapunov vectors in the same column. The Lyapunov vectors represent the principal directions in which errors in the sampling algorithm influence the sampled image. Notice that the principle Lyapunov vectors morph the shape of the sample without destroying image fidelity, whereas the lower Lyapunov vectors destroy image structure. This is consistent with our claim that errors propagate the image tangent to the data manifold.

where $t \sim U(\sigma_{\min}, T)$, $x_0 \sim p_{\text{data}}$ and $\epsilon \sim \mathcal{N}(0, I)$. Once again we note that in the continuous setting we have $\alpha_t = \exp\left(-\int_0^t \beta_s ds\right)$. Training is done in batches of 64 for 30 epochs. The backward process consists of 4000 steps, generating a trajectory of (11) from time $T = 0.9$ down to $\Delta = T/4000$. We use an Adam optimizer with learning rate 2e-5.

Once trained, the score approximation is given by $s_\theta(x, t) = \frac{1}{\sqrt{1-\alpha_t}}\epsilon_\theta(x, t)$. The neural network $\epsilon_\theta$ is a U-Net, that was implemented in PyTorch by [59]. The U-Net consists of two down-sampling stages, one mid-level stage, and two up-sampling stages, where the $28 \times 28$ image is down-sampled to an array of $7 \times 7$ images and up-sampled again. Each downsampling stage consists of two ResNet layers with SiLU nonlinearity and an Attention layer. The mid-level consists of a ResNet layer followed by an Attention layer followed again by a ResNet layer, before up-sampling in a symmetric fashion.

### G.4   CIFAR-10

Our perturbation experiments on the CIFAR-10 data distribution also confirm the robustness of the support property exhibited by score-based diffusion models. In Figure 6 (left), we show images sampled by a pretrained generative model from [53] Github. On the right hand side of Figure 6, we show images generated by the same model with a score perturbation of size 0.1 ($L_\infty$ norm) added to $F_t$ for each $t$. These samples look visually no different and produce similar likelihood scores, $\approx 3.7$ bits/dim, compared to the predicted samples using the original pretrained score model even for perturbation size up to 1. As expected, this behavior of robustness of the support is reproduced with any stable time-integration scheme, e.g., using predictor-corrector or probability flow ODEs.

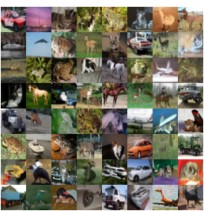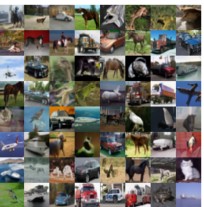

Figure 6: Left: images predicted by a pre-trained score-generative model by Song et al 2021 [Github link] on CIFAR-10 training images. Right: Predicted images by the model when a size 0.1 perturbation is added to the score vector field.

### G.5 Conditional flow matching

So far, all our numerical experiments were carried out with diffusion models. Here we compare the robustness of the support across other conceptually different generative models. Specifically, we consider experiments with conditional flow matching variants [38, 37] and stochastic interpolants [3], and all our experiments are based on the implementation by the `TorchCFM` package Github. At their core, these dynamical generative models interpolate samples from a source density $\rho_0$ and samples from the target $p_{\text{data}}$. For instance, a variance-preserving interpolation is used in stochastic interpolants [3] and a straight line interpolation is proposed in rectified flow sampling [38]. In these generative models, a stochastic path such as $X_t = (1 - t)X_0 + tX_1 + \sigma(t)\xi_t$, with $X_0 \sim \rho_0$, $X_1 \sim p_{\text{data}}$ is predetermined, while the probability flow path is computed. This is in contrast to SGMs, where the probability path is predetermined for the reverse process by choice of the forward process. The score approximation is performed for Brownian/OU paths in SGMs, while for other paths in flow matching. Thus, it is natural to ask if the learned dynamics for these different probability paths also possess the robustness property.

**Less robust flow matching models.** Following `TorchCFM` tutorials [56], we learn vector fields $v_t$ with an MLP and 256 training samples per epoch from the 2 moons data distribution. The generated probability density is quite accurate for all of these models. In Figure 7 (top left), we show the generated density from Optimal Transport-Conditional Flow matching [56]. Next, we add a perturbation of size 0.5 and 1.0 in the $L^\infty$ norm to the learned vector field $v_t$. The predicted densities for OT-CFM (first row) and stochastic interpolants (third row) seem to show the most robustness to the support, while for non-rectified flow matching the densities do not seem to exhibit robustness of the support, in comparison. Visually, all of these models seem to be less robust (c.f. Figure 1) than score-based diffusions. It is noteworthy that this is not due to the effect of the noise in the diffusion process, as the same robustness is visible even for deterministic time-integration (probability flow ODEs) using the scores. Thus, the robustness seems to be dynamical, with different dynamics on probability space and the loss function/formulation together dictating specific dynamics on sample space.

To understand the effect of the dynamics further, we compute the LVs and LEs as before using an iterative QR algorithm. Recall that the LEs are recovered as the time-average of the log diagonal elements of $R_t$ (see section 4 of the main paper). We observe that some paths (i.e., with non-zero probability with respect to the source distribution) may have positive leading LEs, while SGMs were always observed to have stable LEs. We take the source density to be 8 Gaussians, but essentially similar results were obtained with a standard Gaussian source density.

In Figure 8, we show the leading LV (in blue) calculated for three different GMs in the top row. Also plotted is the score of the approximate 2 moons density (shown in red) in each case. The model OT-CFM seems to be most consistent with Theorem 4.3, showing most orthogonality with the score, or alignment, among the flow-matching models, but much less compared with diffusion models. To quantify the alignment, we plot the histogram of the absolute value of the dot product between the normalized score vectors. The generative model using optimal transport appears to have the best alignment since the histogram has a faster decay and a sharper peak at 0 (orthogonality between the score and the leading LV). Although Theorem 4.3 only proves that the orthogonality is a sufficient condition for the robustness of the support, it seems to agree qualitatively with the observations in Figure 7. The most aligned model, being OT-CFM, also exhibits most stability of the predicted support to perturbations. Moreover, none of these models are as robust or as aligned as diffusion

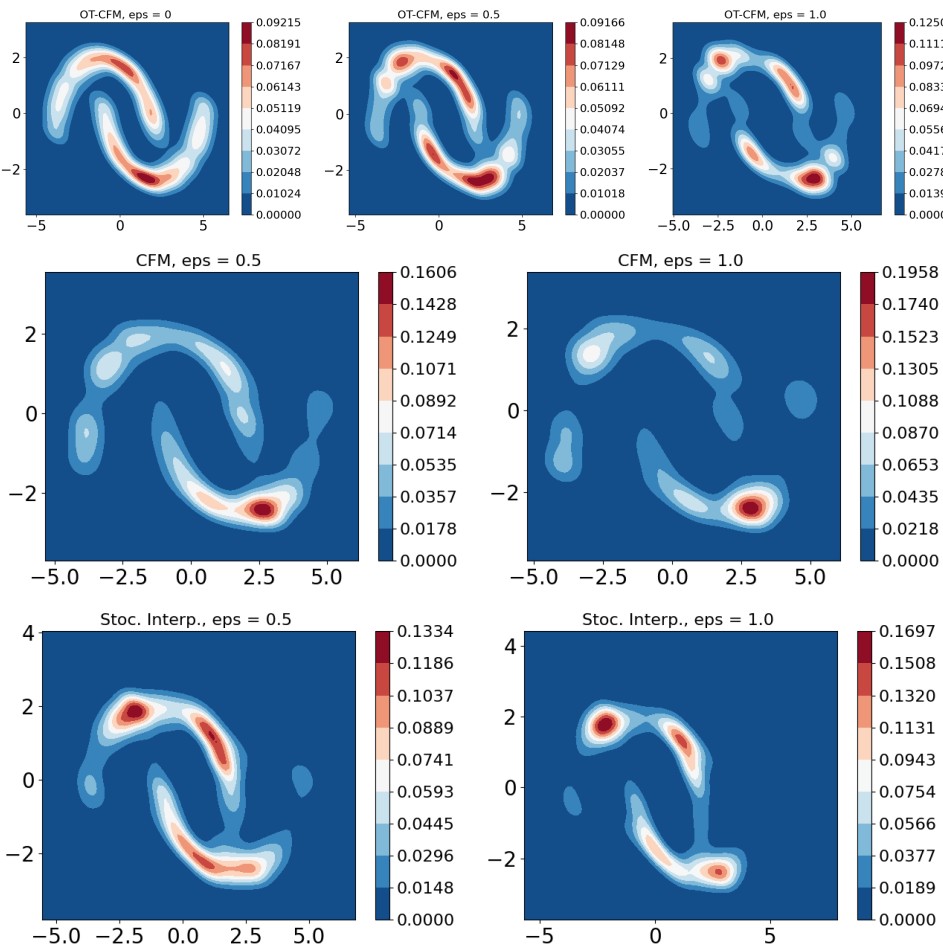

Figure 7: Top row: (Left) Two moons data distribution generated by an optimal transport-conditional flow matching (OT-CFM) algorithm [56]. OT-CFM dynamics perturbed by errors in the vector field of $L^\infty$ norm 0.5 (center) and 1 (right). Middle row: densities predicted by non-rectified flow matching model with perturbations of size 0.5 (left) and 1.0 (right). Bottom: densities predicted by perturbed stochastic interpolant models.

models for the same target. These interesting results can open up avenues to pinpoint the most prevalent cause of robustness or lack thereof of the support. Furthermore, our results can be a starting point to understanding the deep connection between the dynamics on sample space that leads to robustness and the dynamics on probability space (which does not uniquely determine the sample space dynamics).

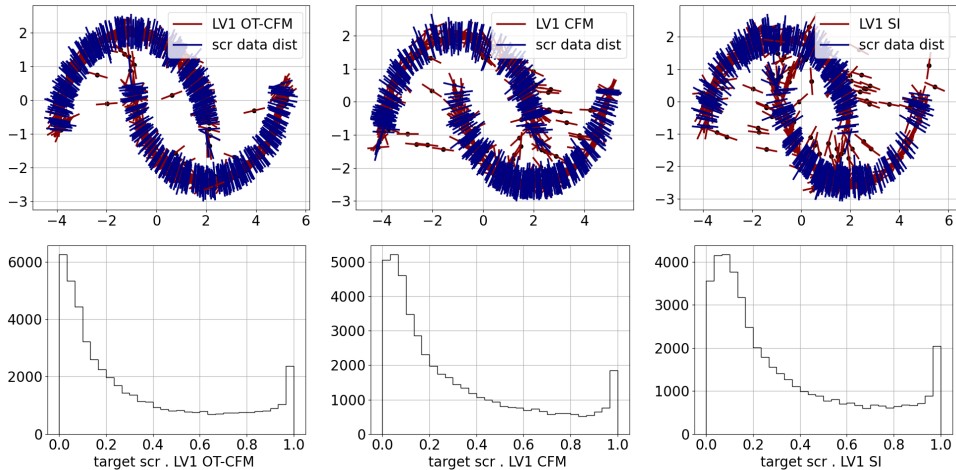

Figure 8: Top row: the target score vector field (blue) and the top LV (red) computed using unperturbed GMs: OT-CFM (left), CFM (center) and stochastic interpolant (right). Bottom row: the histograms of the dot products (absolute value) between the normalized target score and the leading LV (red) over 40,000 points. We see that the stochastic interpolant model and CFM are less aligned than OT-CFM according to our definition in this case.

