# OpenReview forum: "When and how can inexact generative models still sample from the data manifold?"
_NeurIPS.cc/2025/Conference — NeurIPS 2025 poster_

### Official Review · Reviewer_gbmB · 2025-06-26

**Clarity:** 3
**Significance:** 4
**Originality:** 4
**Rating:** 5
**Confidence:** 2

**Summary:**

This paper explores the surprising empirical observation that generative models GMs often generate samples that remain close to the support of the data distribution—even when the learned score function is inaccurate. The authors investigate this phenomenon through a dynamical systems lens, modeling the generative process as a random dynamical system and conducting a perturbation analysis on both the probability flows and sample paths.

**Questions:**

The current theory provides sufficient conditions for robustness. Are there indications of necessary conditions, or empirical observations of counterexamples that satisfy some but not all conditions?

Could the alignment property be enforced or encouraged during training via a regularization term?

Your results are framed for infinitesimal or small perturbations. How does the theory behave when learning errors are non-negligible?

**Ethical Concerns:**

["NO or VERY MINOR ethics concerns only"]

**Final Justification:**

The author has addressed concerns in the review. Recommend an accept.

**Limitations:**

Yes

**Quality:**

4

**Strengths And Weaknesses:**

Strengths
- this is an excellent high-quality paper with deep general insights into the robustness and behavior of the main generative models used today.
- the results are extremely general, applying to all practical implementations of GMs
- the results are clear, extremely significant, and original
- theoretical analysis is rigorous, with careful derivation of perturbation response fields and a novel use of finite-time Lyapunov theory.
- sufficient conditions for support robustness are justified and grounded in both empirical observation and mathematical derivation.
- The paper is well-structured, guiding the reader through a difficult topic with illustrative examples and clear definitions.
- It connects geometry, dynamical systems, and deep generative models in a novel way that could influence future design and evaluation.

weaknesses:
- Concrete suggestions to improve to GMs are not mentioned, even vague suggestions would be welcome.

---

> ### Author Rebuttal · Authors · 2025-07-31
>
> We really appreciate all your keen observations about our paper, the great questions and your encouraging review! In line with your comments, we believe that connecting the dynamics of GMs to the geometry of the data manifold will help us improve both theory and algorithms for GMs. One suggestion we have (and are currently working on) is solve control problems to fine-tuning GMs to ensure specific properties, as we mention in the response to Reviewers YsYs and 678s. Here are a few ideas:
>
>    1. **Studying guided diffusions**: we can take our dynamical systems approach to understand guided SGMs, where an additive drift term changes the target distribution. Apart from analysis of the theory of guided diffusions (how to use them for Bayesian posterior sampling etc), there remain many open questions on how to use guidance for rare event sampling (guiding toward low-probability regions) and for unlearning (guiding away from certain regions of the target support). Taking the dynamical systems approach in such problems will provide a new perspective compared to existing optimal control or learning theoretic approaches to both understand and improve guidance. We believe that in viewing guidance as a control problem (please see responses to Reviewers YsYs and 678s), the most sensitive subspaces we define here can act as a good basis to reduce the dimension of the desired control. Therefore, the dimension of the control or fine-tuning optimization will reduce to the intrinsic dimension as opposed to the ambient dimension, thus also making learning algorithms for such controls feasible.
>
>    2. **Transfer operators**: while this discussion is outside the scope of the paper, an operator theoretic view of the dynamics, via Koopman or transfer operators, is also underexplored in the GM literature. The transfer operator gives the evolution of probability densities under the dynamics. This means that when we consider perturbations to transfer operators due to errors in the dynamics, we can obtain fine-grained control over errors in the predicted probability density. In the computational dynamics literature, there have recently been exciting developments in scalable spectral approximations of these transfer operators. It would be very impactful to consider how the availability of approximate transfer operators can help us build GMs with controlled errors.
>
>
> ## Necessary conditions
>
> You are absolutely correct that the current theory only proves sufficient conditions for robustness of the support. Using the recursion for the score (like Eq. 4) however, we can give a partial answer to your question about necessary conditions. Consider a generative model that expands sample space volume in $(TM)^\perp$, that is, orthogonal to the data manifold. In effect, this amounts to the opposite of alignment: the least stable Lyapunov vectors are orthogonal to the data manifold at the boundary. In addition, if these directions are expanding, that is, $\mathrm{det}(\prod_t R_t) > 1,$ then, any small errors will cause samples to shift away from the data manifold. This will necessarily lead to lack of robustness of the support. Therefore, one necessary condition is not having complete misalignment or orthogonality of $E^d_t$ and $TM.$
>
> Thank you for the very interesting question on examples violating the sufficient conditions of 4.3. We have one such example in Figures 5-6 of the Appendix. Curiously, here, we observe that the sufficient conditions i and ii hold while iii on the cross-derivatives of the drift terms in a conditional flow-matching model is not necessarily satisfied. In this setup, we find that alignment only partially holds, as Figure 6 (left) shows. Your question leads us to ask if having small cross-derivatives (see also our responses to Reviewer YsYs) and a huge attractive force toward the target support is necessary for alignment. Unfortunately, our results neither support nor disagree with this.
>
> ## Fine-tuning to ensure alignment
>
> We believe that alignment can be enforced with a regularization term, as you suggest, or as an additional objective function to fine-tune pretrained GMs (this is our current work in progress). The main idea (also in the response to Reviewer YsYs) is as follows. For alignment to hold, using the proof of Theorem 4.3, the score component along the most sensitive subspace, $s_t E^d_t,$ must decrease with $t$ and become arbitrarily small as $t \to \tau$ on the boundary of $M.$ Since $s_t E^d_t$ satisfies a recursive relationship, as we show in Eq. 3, we can add the recursion to the usual score-matching or flow-matching loss to be minimized: $\sum_t \|s_{t+1} E^d_{t+1} - s_t E^d_t R_t^{-1} - w_t E^d_t R_t^{-1} \|^2.$ Additionally, since we want to minimize $s_\tau E^d_\tau$ to produce alignment, we can add its norm as a regularizer. Since both these terms only depend on the dynamics (solution map $F_t$) and are computable only using the linear algebraic calculations we describe in section 3, the loss function will have a feasible computation that would retain automatic differentiability. Moreover, since the score recursion essentially captures the fact that the dynamics $F^t$ acts as a transport map between the source and target distributions, perhaps this can be used to isolation, without needing the score-matching or flow-matching type loss functions. We believe this idea amounts to knowledge distillation or self-distillation for GMs but derived in a more general context, and we will explore this connection in our future work.
>
> ## Non-negligible perturbations
>
> When learning errors are non-negligible, none of the linear perturbation analysis we carry out in our present paper applies! Nevertheless, viewing the generating process as a random/nonautonomous dynamical system is still valuable in the following ways.
>
> 1.  Under large perturbations, we can treat the incorrect generating process as a completely different dynamical system, say $G_t$. Now, when can $F_t$ (original dynamics) and $G_t$, which are completely different transformations, have the same final density? Since $F_t$ and $G_t$ are different, so are their transfer/Frobenius-Perron operators, say, $\mathcal{L}_t$ and
> $\mathcal{P}_t$ respectively. Now, since we know the approximate spectral properties (using standard operator approximation methods) of $\mathcal{L}^\tau$, we can reproduce this approximation on a finite-dimensional space for $\mathcal{P}^\tau$. In other words, even though the one-step dynamics and the corresponding probability evolutions do not match, we can use the final transfer operator approximation to train a new dynamical system with the same finite-dimensional approximation.
>
> 2. In the above, we have not utilized perturbation theory of operators. Control theory however still applies. Again, let the incorrect generative model be $G_t$ and $F_t$ be the true model that is unknown but produces the correct probability density. We can think of using orbits of $G_t$ the incorrect model with the learning errors to recover the ground truth $F_t$ even if $G_t$ is not a small perturbation. For instance, consider an invertible coordinate transformation $\Phi$ (an encoder or lift represented by a neural network) such that $F_t = \Phi \circ G_t \circ \Phi^{-1}$. Given the desired properties, say alignment that are present in $F_t$, how can we ensure that these hold for $G_t$, the given dynamics? This question can be theoretically analyzed via techniques used to study conjugacies (although these typically consider $\Phi $ to be a small perturbation of the identity) in dynamical systems. Practically, we can consider the problem of training a neural network $\Phi$ to effect desired properties on $G_t$ using only samples from $F^\tau$. These are preliminary ideas that we hope to explore in our future work.
>
> Please let us know if you would like additional elaboration on any of the answers above! We are extremely grateful to you once again for the thought-provoking questions that can spur future efforts in this direction.

---

> > ### Author Response · Authors · 2025-08-06
> > **Request to read rebuttal**
> >
> > Dear Reviewer,
> >
> > Since we are close to the end of the rebuttal period, we request you to spend some time reading our responses, so that we will have time to provide any additional clarification. We acknowledge your thorough initial review and really appreciate the time it must have taken -- sorry for asking for more time!
> > Thank you very much and hope to hear from you!

---

> > ### Comment · Reviewer_gbmB · 2025-08-06
> >
> > The reviewer thanks the authors for their excellent response. However, while the contributions are novel and impactful, the methods used are applying dynamical systems theory to this diffusion context.
> >
> > The reservation this reviewer has now, after some further consideration is the doubts of its accessibility to ML practitioners more widely. The material is some quite dense dynamical systems theory that may be hard to parse. Furthermore practical suggestions have not been solidified.

---

> > > ### Author Response · Authors · 2025-08-07
> > >
> > > Thank you very much for your comment about the novelty and impact of our paper. In response to other reviewers as well, we have revised how we introduce tangent space dynamics and the concept of alignment. We hope that this revision, along with bringing some of the details from Appendix B and C into the main text, will greatly improve the clarity of the paper. We definitely acknowledge that the broader ML community may not have prior exposure to dynamical systems theory. At the same time, the results in our paper indicate that the theory can provide a powerful tool to interpret, understand, improve, and control GMs. We will add a paragraph (to replace some interpretation of the Theorem), similar to the broader impacts/contributions in the introduction, that describes
> > > i) how to verify alignment (please see response to Reviewer YsYs) in practice, and ii) how to bring about alignment and hence robustness in practice (as in the response to you above). Despite being brief, we hope that this addition can lead to practical algorithms in future work for alignment. We view this paper as the introduction to the idea of alignment and how it leads to improved rigorous understanding of the errors made by GMs. The next step would be to control those errors!
> > > Please let us know if you have any other suggestions for revisions, and we really appreciate all your questions and suggestions so far. Thank you once again!

---

### Official Review · Reviewer_Ysys · 2025-06-28

**Clarity:** 1
**Significance:** 2
**Originality:** 3
**Rating:** 4
**Confidence:** 2

**Summary:**

This work attempts to theoretically explain the robust performance of generative models in the presence of approximation error in the score function. The main message of the paper is the following: even under the approximation error, the generated samples drift only in the directions along the support, not away from it. As a result, the generated samples are not too different from the target samples. The main analysis of the work relies on the perturbation theory and analyzes the properties of the top Lyapunov vector (the most sensitive perturbation direction). I think this is an important problem to study and requires future research. However, I find the presentation of the paper extremely hard to follow. I elaborate on this in the later section.

**Questions:**

- What is the main implication of Theorem 4.3? It would be helpful if the authors could provide more figures with explanations.

- Line 275: What is the meaning of " aligned and convergent generative model"? An explicit definition will be helpful to the readers.

- Also, how to verify the alignment of a model in practice?

- Lemma 4.4: What is the meaning of regular? An explicit definition is "alignment property," and "regular" is helpful.

-

line 311: "An SGM typically satisfies this condition.":  Is that trivial? A short argument might be helpful.

**Ethical Concerns:**

["NO or VERY MINOR ethics concerns only"]

**Final Justification:**

My doubts have been clarified.

**Limitations:**

As of now, the scope of the work is very theoretical and limited. Further experimental verifications are necessary to gauge the impact of the work.

**Quality:**

2

**Strengths And Weaknesses:**

# Strength

The paper addresses an important problem that might broaden our theoretical understanding of the contemporary generative models. The paper introduces a novel mathematical framework for analyzing the dynamics of the generated samples using perturbation theory. I am not an expert in perturbation theory, but it seems that the present framework might be helpful in future research in this direction.

# Weakness
I find the presentation of the paper extremely hard ot follow.

- The paper is very notation-heavy, and it is hard to follow the claims for a non-expert reader. I encourage the authors to add some sort of table for notations.

- Line 180 - 185: The phenomenon regarding the top Lyapunov vector explained in the aforementioned lines is not quite clear. It will be helpful if the authors provide more figures to explain the details of the dynamics. More importantly, definite examples supporting the claims would be helpful for the readers. For example, verifying the claims in simple settings, such as the DDP model with a mixture of Gaussian distributions as the target distribution,  is encouraged for better readability.

- The main results of the paper are hard to parse. For example, the implication of Proposition 4.1 is not clear. Also, what is "margin" when you write $c = O(1/ margin)$ in Line 213? I could not find any definition.

---

> ### Author Rebuttal · Authors · 2025-07-30
>
> Thank you for taking the time to improve the readability and clarity of the paper! Below we elucidate all the concepts and terms that were previously unclear, and we will also modify the relevant parts of the paper in the revision -- thank you once again. Please let us know if there is still any misunderstanding in our response, and we are happy to try again.
>
> ## Intuitive explanation of top Lyapunov vectors
> First we begin with what a Lyapunov vector is in our context of a finite-time nonautonomous dynamical system (nonautonomous means that the transformation at time $t,$ denoted $F_t,$ is itself a function of time). As described in the paper, the eigenvectors of $(\nabla F^t)(\nabla F^t)^\top$ or equivalently the left singular vectors of $\nabla F^t$ are the finite-time Lyapunov vectors we use. Please see the following references: Young J Physics 2013, Ginelli et al J. Physics, 2013, for the theory of Lyapunov vectors (in the time-asymptotic case), in addition to the references cited in the paper.
>
> To understand what these left singular vectors mean, we can consider any unit vector, say, $y_0 \in \mathbb{R}^D$ and its evolution under the linearized dynamics (please also see the response to a similar question by Reviewer 678s) that dictates how an infinitesimal perturbation to the initial sample, say $x_0,$ propagates through $F^t,$ $y_t:= \nabla F^t(x_0)  y_0.$ Now, the unit vector $y_t/\|y_t\|$ and the norm $\|y_t\|$ determine the direction and norm of the growth/decay of the perturbation applied to $x_0$ in the direction of $y_0.$ By definition of singular values and singular vectors, the $y_0$ that maximizes the stretch by $\nabla F^t$ is its singular vector corresponding to its top singular value. There are $D$-many (possibly non-distinct) singular values and their logarithms divided by $t$ are referred to as finite-time Lyapunov exponents. The constructive definition of the most sensitive subspaces in section 3 of the paper is an approximate numerically stable algorithm (similar to QR iteration for computing eigenvectors) for computing the top $d$ ($\leq D$) Lyapunov vectors. Thank you very much for asking this question and suggesting further numerical experiments to illustrate these concepts. Currently we show the top Lyapunov vector (the leading direction of infinitesimal perturbation growth) for the 2D generative model in Figure 1, and we have 2 other two-dimensional examples in Appendix G.2. We plan to add one more 2D example as you suggested, in which the target distribution will be supported on the circle, where it will be easy to visualize both the leading Lyapunov vector as well as alignment, which we describe next (please see response to Reviewer qRyA).
>
> ## Intuitive explanation of the concept of alignment and its numerical verification
>
> Alignment means that the most sensitive subspace (the top $d$-Lyapunov vectors defined above) is parallel to the tangent space of the data manifold. As we discuss above, the tangent space of the $d$ data manifold ($M$) is a subspace of $\mathbb{R}^D$ at each point. When the two subspaces, the most sensitive and the tangent to $M$ at each point on the boundary, this leads to the generative model learning the support, and this is our main result (Theorem 4.3).
>
> The most sensitive subspace refers to directions of maximum error propagation in the dynamics $F^t.$ Therefore, for a sample at time $\tau,$ an incorrect dynamics leads to shifts along this subspace. When this subspace is the same as $TM$ infinitesimal errors cause perturbations along $M$, but not away from it. This is what Figure 1 illustrates, where there is alignment. When alignment is not present (for example, with CFM in Figures 5 and 6 of the Appendix), we see that learning errors in the drift term can cause samples to move away from the data manifold. One way to make this argument more quantitative is via Proposition 4.1, which we describe next.
>
> ## Definition of classification margin and why margin does not change under alignment
>
> For binary classification by a hyperplane classifier, margin is the minimum distance between the training points and the classifying hyperplane. The relationship between the margin and generalization of a binary classifier is a well-known part of statistical learning (see e.g., Mohri et al 2018). In our context, we leverage these classical results for a kernel binary classifier to quantitatively describe how an incorrect model can still learn the support.
>
> Learning the support of a probability distribution can be cast as a problem of a one-class binary classification: given only positive samples (that is, points from the support), learn a binary function that evaluates to -1 when samples are out-of-support. If we use the generated samples as positive training samples, can we learn the correct classifier for the target distribution? The main novelty of the proof is the recognition that alignment is the condition under which the margin of a classifier learned with true samples and one with samples from an incorrect generative model coincide. This leads to identical generalization bounds and thus we obtain quantitative estimates on how well an aligned generative model can learn the support. Thank you! We will add an abridged version of this response to the revision.
>
> The incorrect generative model is only a small perturbation of an ideal GM. This is what is captured by **convergence**. We refer to the existing literature on convergence of GMs. We consider a convergence result from Lee, Lu and Tan in COLT 2023. Suppose the score is learned with an L2 error $\mathcal{\epsilon}$. Then, they show that the W2 distance between the predicted density, and the target is $\mathcal{O}(\epsilon^{1/18})$. We use this result and optimal transport theory combined with margin-based classification bounds (above) to obtain Proposition 4.1. Please see Appendix D for more. We plan to add the definition of convergent GMs to the main text.
>
> ## Implications of Theorem 4.3: sufficient conditions for alignment
>
> 1. When alignment holds, the most sensitive subspaces directly give the tangent basis to the data manifold. This construction exploits the dynamics of generative models and thus provides an efficient way to learn the data manifold. This point appears in the paper after the proof of 4.3.
>
> 2. The recursive equation for the score component along the data manifold, given in Eq. 4, provides mathematical insight into how errors in generative models propagate to the predicted density to either produce or preclude alignment. This equation (see also our answers to Reviewer qRyA) is how the dynamics affects alignment. For example, it says that positive finite-time LEs, i.e., expanding sample space volume, can make it easier for alignment to hold. Thus, this can lead to a principled basis on which to distinguish between the many existing GM algorithms.
>
> 3. We can rederive Eq. 4 and control the left hand side by replacing $F_t$ with a deliberately perturbed generative model, e.g., $F_t \to F_t + u_t,$ a controlled dynamical system. Minimizing the new left hand side over all $u_t$ will produce alignment in the controlled generative model. The same approach can produce another desired property, e.g., for unlearning. Thus, the techniques used in 4.3 are applicable for controlling generative models, going beyond alignment (please see our response to Reviewers 678s and gbmB).
>
> **verifying alignment in practice**: this is an excellent question! In low-dimensional examples, such as Figure 1 of the main text and Figures 5 and 6 of the Appendix, we can directly visualize the tangent spaces to the data manifold and the most sensitive subspaces, and therefore alignment. For high-dimensional examples, we have developed two indirect ways of verifying alignment:
>
> 1. In Fig 2 of the main paper and Fig 3 of the Appx, the indirect verification is based on moving along the most sensitive subspace (easy to compute using their definitions, see section 3) and check if we still obtain a sample on M. If yes, alignment holds.
>
> 2. When M is low-dimensional (see answers to Reviewers 678y and qRYa), the target score is orthogonal to $TM$. Thus, alignment essentially means orthogonality of the score and the most sensitive subspaces. This is what is used in Figs 2 and 5-6 in the Appendix. This approach is scalable.
>
> ## Definition of regularity of alignment
>
> Regularity of a property, across different areas of math, means stability of the property under perturbations. In our context, we ask whether a GM with alignment will also retain alignment under perturbations. The answer is yes, as shown in the paper. This is helpful in practice because if we build a GM with alignment, we do not need to worry that even a smooth perturbation will ruin the alignment.
>
> ## Why do cross-derivatives have small norm?
>
> Thank you for this great observation whose explanation we have omitted on line 311. In an SGM, the drift term is the same as the score. Therefore, near time $\tau,$ the drift vector field has the same structure as the target score. Now, the target score is undefined at $\tau$ and has uniformly large values pointing away from M under the manifold hypothesis when $t$ is close to $\tau$. Because of this, the score component orthogonal to M does not change drastically in the direction of data manifold (along $E^d_t$), since it takes on uniformly large values. Similarly, the score component tangent to the data manifold is changing more rapidly in the tangent direction, indicating the fine-grained features of the target probability density, when compared to in the orthogonal direction. This is because the density outside the support is zero uniformly. You are right that this message will be enhanced with the mixture of Gaussians (vonMises distributions) example that you suggested.
>
> Thank you again for your time, and we would be very grateful if you could reassess our contributions in light of our answers and revisions.

---

> > ### Comment · Reviewer_Ysys · 2025-08-02
> >
> > Thank you for the clarification. I suggest that adding the above explanations in the paper would significantly increase the clarity of the work. I have raised my score.

---

> > > ### Author Response · Authors · 2025-08-02
> > > **Thank you!**
> > >
> > > Thank you very much for reading our responses and reassessing our work! We will certainly make all the revisions we have promised above in our paper and appendix, in response to the points you have raised. Thank you very much once again for all your time and efforts to improve the clarity of our paper.

---

### Official Review · Reviewer_qRyA · 2025-07-03

**Clarity:** 2
**Significance:** 3
**Originality:** 3
**Rating:** 4
**Confidence:** 3

**Summary:**

In this paper, authors study the robust behavior of dynamical generative models.

Authors first show that even with erroneous perturbation, the infinitesimal change of the target density is only supported where the target density is supported. To explore the manifold hypothesis, the authors introduce the notion of alignment: the principal directions of sample deformation—captured by the leading Lyapunov vectors—are orthogonal to the target score at the boundary of the data manifold. A sufficient condition for alignment is also derived for generative models.

Both analyses explain the robustness of dynamical generative models and why they can reliably produce samples on the target manifold even with statistical estimation errors.

Some empirical results are performed on toy and real-world datasets, validating the theoretical claim.

**Questions:**

If the author could kindly answer the following questions, it would greatly helpful:

- > Intuitively this means that the generating dynamics may include an expansive phase and the vector field vt is a
 uniform “attracting force” at the end.

Could authors explain how this intuition is derived from the sufficient conditions listed in Theorem 4.3? Specifically, it is not clear to me which part of the sufficient condition implies that we could have an expansive phase.

- What does the condition (ii) mean in Theorem 4.3? Could we check if this condition is satisfied in Figure 1?

**Ethical Concerns:**

["NO or VERY MINOR ethics concerns only"]

**Final Justification:**

If the author could incorporate more intuitive explanations into the revision, this paper would be stronger. All my confusions are addressed, and after reading other reviews, I am now more confident that this is a technically sound paper.

**Limitations:**

Yes.

**Quality:**

3

**Strengths And Weaknesses:**

Strengths:

1. The manifold hypothesis (Song and Emron., 2019, [45, 14] in the manuscript) is an important conjecture and is believed to be the reason that dynamical generative model work so well comparing to other generative methods (e.g. GAN). Mathematical analysis on this subject will have significant impact on down-stream applications.
2. Authors' analysis assumes a very generic framework of dynamical generative models; thus, the results are applicable to a wide variety of drift based generative models (such as diffusion models and CFM).
3. The sufficient conditions of manifold alignment theorem have intuitive interpretations and can be computationally validated even on high-dimensional data (Figure 2).

Song and Emron, Generative Modeling by Estimating Gradients of the Data Distribution, 2019

Weaknesses and Comments:

Overall, this paper is technically dense and could benefit from additional examples illustrating these mathematical concepts.

The first result (Section 3) is not very well explained. Authors could expand the paragraph after eq. 2, explaining the result and its implications in a more intuitive manner (similar to line 274 - 288). For example, why does the eq. 2 show that generative model sticks to the target support? It is not immediately clear as the rho_tau(x) in eq.2 is not target density p_data.

Some elements in the sufficient conditions in Theorem 4.3 are very abstract, (e.g., nabla_t,d v_t,d), and are not easily observable from Figure 1. Could authors create a new plot to show how do these quantities behave on the boundary of the manifold?

After Theorem 4.3. I recommend introduce some toy examples (e.g. diffusion generative model learning a target density on a sphere) and show the validity of the sufficient conditions using this simple example.

To a general reader who may not have extensive experience on dynamical generative model, some terminologies may be hard to understand. For example, "compressive", "expansive phase", "superstable", "non-singular target density" etc.

---

> ### Author Rebuttal · Authors · 2025-07-30
>
> We thank the reviewer for taking the time to provide us with many suggestions to improve the clarity and presentation of the paper.
>
> ## Intuitive explanation for statistical response-based robustness
>
> As the reviewer has pointed out, we have omitted an intuitive explanation for why Eq. 2 (on statistical response) implies the robustness of the support. We have now added the following explanation to the revision. Before that, however, we would like to point out that $\rho_\tau$ is indeed the density of the target distribution, $p_\mathrm{data},$ if $p_\mathrm{data}$ has a density (with respect to Lebesgue measure, i.e., it is absolutely continuous or not singular). When $p_\mathrm{data}$ does not have a density (e.g., if a discrete probability distribution or a continuous distribution supported on a low-dimensional manifold), then $\rho_\tau$ is an approximate probability density of $p_\mathrm{data}.$ The latter case is best exemplified in a diffusion model, where a time-integrating the reverse process from 0 to $\tau - \delta t$, i.e., close to the final time (just before the score blows up or is undefined) still results in a probability density: this probability density is the density of the distribution $p_\mathrm{data}$ convolved with a  Gaussian of small variance and mean 0.
>
> Having clarified the meaning of $\rho_\tau$ as the density of the target or the approximate density of the target, we now interpret the implications of Eq. 2 for robustness of the support. In Eq. 2, the density $\rho_{\tau, \epsilon}$ is the density obtained from an incorrect GM in which the magnitude of learning errors is $\epsilon.$ This equation compares the incorrect density $\rho_{\tau, \epsilon}$ with $\rho_\tau,$ the target density produced by a perfect GM. For small $\epsilon,$ we can rewrite the left hand side as $(\rho_{\tau, \epsilon} - \rho_\tau)/\epsilon$ and the right hand side is the product of a scalar function, say, $\alpha(x) := \sum_{t=0}^{\tau -1} ( \mathrm{div}(\chi_t) + \chi_t \cdot s_t) \circ F_t^{-1} \cdots F_{\tau-1}^{-1}(x),$ and $\rho_\tau.$
> Hence, we can rewrite Eq. 2 as $\rho_{\tau, \epsilon}(x) - \rho_\tau(x) = \epsilon  \alpha(x)  \rho_\tau(x) + \mathcal{O}(\epsilon^2).$ There are a number of implications of this result:
>
>  1.  The leading order (in $\epsilon$) term is zero where $\rho_\tau$ is zero. As a result, the leading order contribution to the error in the densities, $\rho_{\tau, \epsilon}(x) - \rho_\tau(x),$ is made on the support of the target. Alternatively, at an $x$ outside the support of the target density, the error is $\mathcal{O}(\epsilon^2).$
>  2.  Secondly, the error, $\rho_{\tau, \epsilon}(x) - \rho_\tau(x),$ for a small $\epsilon$ only depends on the unperturbed GM, which has no errors. This is because, in order to evaluate the error at an $x,$ we only need to evaluate the score and the perturbation vector field $\chi$ (and its divergence) along an unperturbed orbit, since the terms involved in $\alpha$ only contain the maps $F_t$ and not $F_{t, \epsilon}.$ Hence, the error in the predicted density only depend on the properties of the original dynamics.
> 3. In regions of high target density (non-negligible $\rho_\tau$), the error depends on the dot product of the perturbation $\chi_t$ with the score functions $s_t$ along the unperturbed orbit. This is intuitive and tells us that a perturbation more aligned with gradients of log density along an orbit will have a greater effect on the predicted density at time $\tau.$
>
> Thank you again for this question, and we will add an abridged version of the response above to the paper after Eq. 2, as you suggest.
>
> **Non-singular targets**: these are target  probability distributions that are not absolutely continuous with respect to Lebesgue (volume) measure in $\mathbb{R}^D,$ the ambient space. We will add a clarification to this and the dynamical systems and perturbation theory related terms (which are explained below) in the main text space permitting, otherwise to the appendix. Thank you for the suggestion!
>
> ## Intuition behind sufficient conditions for robustness; illustrative examples for what derivatives along and away from support mean}
>
> You are absolutely correct that more explanation is needed for the abstractly defined cross-derivatives and how their values impact robustness of the support. We also agree that the illustrations presented for the two-moons example in Figure 1 may be insufficient to observe the evolution of the derivatives of the vector field. Therefore, we will add experiments on a mixture of Gaussians on the circle (since three dimensional vector fields on the 2-sphere may still be hard to visualize) or von Mises distributions. Please see a related answer we wrote for Reviewer Ysys. Here we will focus on providing a textual description to convey the intuition as best as we can.
>
> ## Meaning of cross-derivatives and the local coordinates: explanation for sufficient conditions
>
> First we start with the motivation for the choice of local coordinates. At each point $x,$ we have a tangent space $T_x \mathbb{R}^D$ that consists of all possible directions in which infinitesimal perturbations (to any function) can act. The fundamental idea behind robustness of the support is that perturbations to predicted samples are most significant along the support, i.e., in the subspace $T_x M \subset T_x \mathbb{R}^D,$ where $M$ is the support of the target. When $M$ is of a lower dimension than $D,$ the ambient dimension, we know that the score of the target is aligned with $(T_x M)^\perp.$ Although the target score is technically not defined, we may approximately consider as the target score, the score of $p_\mathrm{data}$ convolved with a Gaussian, as described above. Since the log of this density changes the most in the direction orthogonal to the data manifold, the direction of its gradient (the score) lies in $(T_x M)^\perp.$ The fundamental idea behind our alignment proof is to show that the angle between the score at time $t,$ $s_t,$ and the most sensitive subspace, $E^d_t,$ grows and at the final time, these two vector fields become orthogonal. Thus, $E^d_t$ becomes equivalent to $T_x M,$ which is what alignment means. Because we want to bound the angle between $s_t$ and $E^d_t,$ using local coordinates along $E^d_t$ and $(E^d_t)^\perp$ become a natural choice. Intuitively, this means rotating the Euclidean coordinate system so that now the first $d$ coordinates of a vector in $\mathbb{R}^D$ in this new coordinate system are the components along $E^d_t$ and the last $D-d$ coordinates are the components along $(E^d_t)^\perp.$
>
> Having defined these local coordinates, the derivatives of any function must be accordingly rotated. That is, the rotated coordinate system, say $C:\mathbb{R}^D \to \mathbb{R}^D,$ near an $x,$ is defined with the following rules: $C(0) = x$ and $\partial_i C(y) := dC(y) e_i = (E^d_t)(i)$ where $e_i$ is the $i$th standard basis vector. Here $A(i)$ refers to the $i$th column of a matrix $A.$ With these change of coordinates, what we must prove conditions under which $s_t \cdot E^d_t = \sum_{i=1}^d \partial_i \log \rho_t$ become smaller as $t$ increases. To find these conditions, we use the fact that the score components $\sum_{i=1}^d \partial_i \log \rho_t$ and the dynamics $F_t$ are related because i) $F_t$ evolves the densities $\rho_t$ and ii) $F_t$ also defines the directions $\partial_i \equiv (E^d_t)_i$ recursively.
>
> ## What expansion means and why it helps with alignment
>
> As derived in the paper, we obtain that $\sum_{i=1}^d \partial_i \log \rho_t = s_t E^d_t$ evolves according to the following recursive equation: $\sum_{i=1}^d (\partial_i \log \rho_{t+1})(x_{t+1}) = \sum_{i=1}^d \partial_i \log \rho_t(x_t) \: R_t^{-1} - w_t \: E^d_t \: R_t^{-1}.$ Consider the equation for a sequence of scalars, $p_t$, $p_{t+1} = p_t/R_t - w_t E^d_t/R_t.$
>
> **Expansive and compressive dynamics.** In dynamics, we say $F_t$ is ``expansive'' if $\mathrm{det}(R_t) > 1.$  Intuitively, this means the under the transformation $F_t,$ volumes are expanded (the Jacobian determinant of the transformation gives the contraction or expansion). Similarly, a dynamical system $F_t$ is compressive if $|\mathrm{det}(dF_t)| = \mathrm{det}(R_t) < 1.$
>
> Going back to the equation for the evolution of score components, we can clearly see that the map from $p_t$ to $p_{t+1}$ is contractive when $\mathrm{det}(R_t) > 1.$ Thus, expanding dynamics at each time $t$ leads to alignment. However, we do not observe expansion for most orbits in our numerical experiments with SGMs and CFMs. Thus, we come up with sufficient conditions that allow for contraction, and yet $p_t$ still tends to an arbitrarily small value. To derive such a condition, we impose conditions on the vector $w_t = \mathrm{Tr}(\nabla^2 F_t (\nabla F_t)^{-1})$ and this is what leads to the seemingly unintuitive sufficient conditions ii and iii. The complete proof is in Appendix E. We will revise our explanation in 274-278 to make it more intuitive. We will also add the more easy-to-observe example on the circle, since the tangent and normal spaces are very easy to see.
>
> We have answered all the questions above although not in the order asked. Please let us know if any additional clarification is needed. We are once again very grateful for your thoughtful questions; we request you to kindly reevaluate our work based on the answers above.

---

> > ### Comment · Reviewer_qRyA · 2025-08-02
> > **Thanks for your response**
> >
> > The reviewer thanks authors for their detailed response to our questions and clarifying technical details.
> >
> > I think it would be great if authors could integrate more discussions of intuitions into the main paper. I think many non-theoreticians would also be interested in your results on why generative model stays on the data manifold.
> >
> > I do not have any further questions and look forward to discussing with other reviewers.

---

> > > ### Author Response · Authors · 2025-08-02
> > > **Thank you!**
> > >
> > > Thank you very much for reading our responses and for you comment -- we will surely incorporate the intuitive explanations above (and in our other rebuttals) to your many insightful questions in the main paper. Thank you again for your time and efforts that have led to substantial improvements to our paper!

---

### Official Review · Reviewer_678s · 2025-07-03

**Clarity:** 3
**Significance:** 3
**Originality:** 4
**Rating:** 5
**Confidence:** 2

**Summary:**

This paper investigates a key empirical observation in dynamical generative models, where errors in the learned vector field (score) often shift generated samples along the data manifold but not off it. The authors propose a theoretical explanation for this phenomenon via a dynamical systems perspective, focusing on finite-time Lyapunov vectors.

**Questions:**

1. I am a bit confused about the 'tangent perturbation' at line 120. Does it mean the perturbation along or away from the data manifold? To study the robustness of support, I believe we should consider the perturbation pushing the samples away from the manifold, but 'tangent' sounds like making the samples stick to the manifold. Same question for the last column of Figure 1: what does it mean by shifted 'tangent' to the data manifold? Also would you please clarify what it means by 'perturbations along sample paths' at line 157?

2. At line 68, the author states '... as the perturbation size tends to 0', which looks like indicating an asymptotic result. But I would expect the result like 'as the perturbation size $\epsilon$ is sufficiently small'. Please correct me if I have a misunderstanding.

3. The notation of $u$ defined in section 3 is a bit confusing, as $u$ is a vector field while $u(\rho_\tau)$ is a scalar. I would suggest using different notations.

**Ethical Concerns:**

["NO or VERY MINOR ethics concerns only"]

**Final Justification:**

All my concerns have been addressed, and so I raised my score.

**Limitations:**

Yes.

**Quality:**

4

**Strengths And Weaknesses:**

The paper provides rigorous and detailed mathematical analysis, bridging generative modeling with classical dynamical systems theory, particularly via Lyapunov analysis and transfer operators. However, the paper focuses on sufficient but not necessary conditions, leaving the full characterization of robustness open.

---

> ### Author Rebuttal · Authors · 2025-07-28
>
> We thank the reviewer for these insightful comments and questions. We completely agree that we only provide a sufficient but not necessary condition on the dynamics for the robustness of the support. Rather than a full characterization of robustness, we aimed to introduce two concepts that can have an impact beyond the analysis in the present paper: most sensitive subspace (least stable finite-time Lyapunov subspace) and alignment (the tangency of the most sensitive subspace with the tangent space of the target support). In future work, we anticipate that alignment will be used as an objective function to fine-tune generative models (please see responses to Reviewers YsYs and gbmB on a related question). Since the most sensitive subspaces are defined constructively using an efficient algorithm, they make maximizing the objective function computationally feasible.
>
> Beyond the application to robustness in the present paper, we believe that computable bases for the tangent spaces (which contain errors and control terms) can be useful for optimizing other objective functions, e.g., control-based objective functions in rare event sampling or unlearning. That is, the tangent bases introduced in this paper establish principles for controlling generative models to bias the target distribution for downstream applications, which is a subject of active research in multiple communities.
>
> ## Definition of tangent perturbations
>
> On Line 120, we call the perturbation vector field $\chi_t$  a *tangent perturbation*. This simply means that the additive perturbation $\chi_t$ is a vector field: a function that evaluates to a tangent vector in $T_x \mathbb{R}^D$ at each point $x \in \mathbb{R}^D$. In this context, tangent vector is used in the usual sense of differential geometry; we may think of $\chi_t(x)$ at a point $x \in \mathbb{R}^D$ as a vector in $\mathbb{R}^D$ that represents all possible directions of infinitesimal perturbations (velocity vectors) at $x \in \mathbb{R}^D.$ Most generally, $\chi_t(x),$ will have components both along the tangent spaces to the data manifold as well as orthogonal to it. In other words,  at each $x$ in the support of the target, we have the orthogonal decomposition $T_x \mathbb{R}^D = T_x M \oplus (T_x M)^\perp$ into linear subspaces tangent to and orthogonal to the data manifold, $M$. The tangent space to the data manifold at $x$, denoted $T_x M,$ is a $d$-dimensional linear subspace, where $d < D.$  We have corrected this ambiguity from the use of the word ``tangent space'' for both $T_x \mathbb{R}^D$ and $T_x M$ in the revised version. Thank you for pointing it out!
>
> In Figure 1, we observe that the samples are shifted tangent to the data manifold, $M,$ which is the $d$-dimensional support of the target. This means the following. Consider a sample $x + \epsilon y$, for a point $x$ on the data manifold (support of the target distribution, which is assumed to be a $d$-dimensional manifold) and a small $\epsilon.$ The shift $y$ from $x$ is a tangent vector, and this vector can have components along the manifold (i.e., on $T_x M$) and orthogonal to it (i.e., on $(T_x M)^\perp$). One can write $y \in T_x \mathbb{R}^D$ as an orthogonal decomposition, $y = y_t + y_o,$ where $y_t \in T_x M$ and  $y_o \in (T_x M)^\perp.$ However, interestingly, we find that under alignment, the component $y_o$ of the generated samples of SGMs is negligible. This is what is illustrated in the figure and serves as the motivation for the theory in the paper.
>
> In line 157, we introduce the notion of evolution of infinitesimal perturbations along orbits of $F_t$ (the generating process), which we describe in the remainder of the section. Briefly, using the notation above, we are interested in studying the cumulative effect of infinitesimal perturbations along $\chi_t,$ which captures the error at time $t.$ To understand this, let us fix an initial sample $x_0$ from the source distribution, and consider the original dynamics, $F^t_0$ so that $x_\tau$ is a sample according to the target. Now, starting at the same initial sample, $x_0,$ suppose we apply incorrect (perturbed) dynamics at time 0. That is, we will get $x_1^\epsilon = F_{0, \epsilon}(x_0) = F_0(x_0) + \epsilon \chi_0(x_0) + \mathcal{O}(\epsilon^2) = x_1 + \epsilon \chi_0(x_0) + \mathcal{O}(\epsilon^2).$ Then, we can define a new vector field, say, $u_1,$ which evaluates to $u_1(x_1) = \lim_{\epsilon \to 0} (x_1^\epsilon - x_1)/\epsilon = \chi_0(x_0).$ By definition,  $u_1$ captures the difference between the two sample paths (governed by the original and the incorrect dynamics) at time 1. We can capture the cumulative effect of infinitesimal perturbations ($\epsilon \to 0$) at any time $t$ using the tangent dynamics, as described in the main text of section 3. We will revise the first paragraph of this section to make the motivation clear -- thank you for raising this point!
>
>  ## About infinitesimal perturbations
>
> Thank you for asking about infinitesimal versus non-infinitesimal perturbation size, a question similar to one posed by Reviewer gbmB. In the previous section, we defined the vector field $u_1$ even in the case when $\epsilon$ does not go to 0. However, for the linear perturbation analysis for longer times to hold, in general, we need the limit $\epsilon \to 0.$ This is because we linearize about an orbit (hence we are able to construct dynamics on the tangent space) and for this linearization to hold, the errors in the difference in the orbits must be on the $\mathcal{O}(\epsilon)$. That is, a Taylor expansion for the perturbed orbit about the original must be accurate. Thus, we are restricted to ``small'' and more rigorously infinitesimal perturbations. Thank you for pointing out this clarification, which have added in the revision. When the learning errors are large, perturbation analyses are inapplicable, and we need to take an operator or control theoretic perspective instead, as we discuss in the response to Reviewer gbmB.
>
> ## Notation updates for vector fields and their different interpretations
>
> Vector fields are used in two senses in differential geometry, and both these are applied in section 3. In one sense, we interpret vector fields as functions from $\mathbb{R}^D$ to $\mathbb{R}^D,$ which evaluate to infinitesimal perturbations at each point, as explained above. The second sense is as a linear functional, and it is in this sense that we define the vector field $u$ in Eq. 2. A vector field, say $u$ at each point, gives a direction to take derivatives. That is, it acts as a linear operator that defines how infinitesimal perturbations in the direction of the vector field affect the values of any differentiable function. In other words, if at a point $x,$ the value of a differentiable scalar function, say $f,$ changes along a vector field $u,$ by the amount, $\lim_{\epsilon \to 0} (f(x + \epsilon u(x)) - f(x))/\epsilon.$ This value, called the directional derivative, is also the value of the linear operator: $u(f)(x) := \lim_{\epsilon \to 0} (f(x + \epsilon u(x)) - f(x))/\epsilon.$ But, we agree with the Reviewer that this notation is confusing. We reverted to writing $\nabla f\cdot u,$ with the caveat that $u(f)$ is defined even when $f$ is not differentiable in all directions but only along $u.$ But the existence of $\nabla f$ requires that $f$ is differentiable in all directions.
>
> We have answered all your questions, but we would be very happy to resolve any other issues that you find with the paper or the rebuttal above. Thank you very much once again for your time, and we kindly request you to reevaluate our work based on our responses and revisions above!

---

> > ### Comment · Reviewer_678s · 2025-08-04
> >
> > Thank you for the response. Most of my concerns are addressed, but I am still a bit confused about the 'tangent perturbation', probably because I am not an expert of dfferential geometry. Can you give an example of your $x + \epsilon y$ in the response? For example, if $M$ is the unit sphere in $R^3$ and $x = (1, 0, 0)$. Can you give a tangent vector $y$ explicitly and its components along and orthogonal to $T_x M$? Thanks!

---

> ### Author Response · Authors · 2025-08-04
> **Thank you + tangent perturbation**
>
> Thank you very much for reading our rebuttal!
> Take $y$ to be any unit vector in $\mathbb{R}^3$. That is, the base point $x$ can be perturbed in any direction in $\mathbb{R}^3$, and this is the meaning of a tangent vector: a direction of infinitesimal perturbation. Let us take $y = [a, b, c]$ such that $a^2 + b^2 + c^2 = 1$. That is $y$ is any direction on the unit sphere. Now, at $x = [1,0,0]$, the tangent plane to the sphere ($T_x M$) is parallel to the y-z plane and perpendicular to $x$. So, in this case, the orthogonal decomposition in our response above is $y = y_t + y_o$, where $y_t = [0, b, c]$ and $y_o = [a, 0, 0].$ The component of the perturbation $y$ tangent to $M$ is $[0, b, c]$ while the component of the perturbation orthogonal to $T_x M$ is $[a, 0, 0].$
>
> What this means is we apply a generic perturbation (unfortunately we called this tangent perturbation because it belongs to the tangent space $T_x\mathbb{R}^3$, which is not the same as $T_xM$) along $y = [a,b,c]$ at each time. But, the cumulative effect of this perturbation, captured by the subspace $E^d_\tau$, ($E^2_\tau$, since $d = 2$ in this case) becomes tangent to $M$, under alignment. So if the sample at time $\tau$ is at $x,$ $E^2_\tau$ computed along an orbit that ends up at $x$ is approximately the y-z plane. This is what we call alignment.
> We hope this clarifies all the confusion. If not, we will be happy to try again!
> We really appreciate all your questions and the time you are spending on improving our paper -- thank you.

---

> > ### Comment · Reviewer_678s · 2025-08-06
> >
> > Thank you for the response. I see what you mean: the $tangent$ vector $y$ is not necessarily a vector in the $tangent$ space $T_x M$ at $x$. That's why I get confused. I would suggest using a different name if possible. However, as all my concerns have been addressed, I just raised my score.

---

> > > ### Author Response · Authors · 2025-08-06
> > > **Thank you!**
> > >
> > > Yes, thank you, you are absolutely correct. "Tangent" perturbation does not mean tangency to $M$, rather just that the vector is in the tangent space, $T_x \mathbb{R}^D \neq T_x M$.
> > > Thank you very much for spending the time to seek this clarification, to really understand our work and reevaluate it!
> > > We really appreciate your efforts. Thank you once again!

---

### Note · Authors · 2025-08-11

We thank you all for spending the time to read our paper and rebuttal, and for the many thoughtful questions and comments that have made our work better. Here we summarize changes that address the two main types of comments, about clarity and practical impact:

1. **Clearer/graphical presentation of main results**: we move one of the 2D examples from the appendix to the main text and add another 2D example on the sphere in $\mathbb{R}^3$ to explain
      1. alignment: central idea we introduce that means that the least stable Lyapunov vectors are tangent to the target support,
      1. concepts from the perturbation theory, including $E^d_t$, which is approximately the least stable eigenspace of the matrix $(dF^t) (dF^t)^\top$, and
      1. the meaning of derivatives along $E^d_t$ and $(E^d_t)^\perp$, which appear in the sufficient conditions in Thm 4.3.

2. **Specific practical impact**: As Reviewers suggest, we expand on the practical impact of our results, on how to verify alignment and how to improve robustness of the support. For this, we add text (building on rebuttal and lines 80-86 of the paper) that additionally covers:
     1. **Preserving invariances**:  for singular targets in science, the support is often defined by some invariances (conservation laws) and group symmetries. Our results imply that ensuring alignment yields predicted targets that obey these known physical laws.
     1.   **Controlling GMs**: we can learn control terms to change the distribution predicted by a pretrained GM for applications like unlearning and rare event sampling. Let $x_t$ be an orbit from a pre-trained GM so that $x_\tau \sim p_\mathrm{data}$ but $x_\tau$ is an unwanted sample. We can now learn a dynamic control network $u_t(x)$ so that $y_{t+1} := F_t(x_t + u_t(x_t))$ is a desirable orbit: $y_\tau \sim p_\mathrm{data}$ and is a desirable sample. For a pair of *good* and *bad* training orbits, $y_t$ and $x_t$ respectively, $u_t$ minimizes $\|u_{t+1}(x_{t+1}) - dF_t  u_t(x_t) - (y_{t+1} - x_{t+1})\|^2.$ Further, we need only learn $u_t E^t_d \in \mathbb{R}^d$ (as opposed to $u_t \in \mathbb{R}^D$) at each $t$. We will expand on this brief description to show how to exploit the dynamics of pretrained GMs to control their output samples.

Summary of strengths: all reviewers agree the paper provides original and insightful results on the geometric, probabilistic and dynamic aspects of GMs, improving our understanding and ability to control errors.

---

### Decision · Program_Chairs · 2025-09-17

**Decision:**

Accept (poster)

**Comment:**

In this paper, the authors analyze both stochastic and deterministic dynamical generative models to explain a robustness phenomenon: learning errors in the score or drift field tend to move generated samples along the data manifold rather than away from it. Using perturbation analysis of probability flows and sample paths, they show that the predicted density differs from the target density only on the data manifold. They identify the cause as an alignment between the most sensitive perturbation direction and the tangent space of the manifold, prove a sufficient condition for this alignment, and show that it is efficient to compute. The results apply to different dynamical  generative models, regardless of whether the manifold hypothesis holds, and the theory is validated with a basic experimental setting.

The reviewers acknowledge that the paper offers a significant theoretical analysis that helps explain phenomena previously observed empirically. In particular, the paper:

1. introduces a mathematical framework connecting geometry, dynamical systems, and deep generative models, using perturbation theory to analyze the dynamics, This framework applies to a wide range of models and provides insights into their robust behavior

2. presents theoretical results that appear rigorous and sound, supported by the provided experiments

3. addresses a relevant problem with the potential for significant impact

Some concerns were raised, mainly regarding the clarity and accessibility of the manuscript. While most of these points were addressed in the rebuttal, the paper would benefit from including high-level explanations of the theoretical results, along with visualizations to help build intuition around the key concepts. I therefore recommend acceptance and encourage the authors to consider the feedback and update the final version of the paper accordingly.